# The Green Edge cruise: Investigating the Marginal Ice Zone processes during late spring / early summer to understand the fate of the Arctic phytoplankton bloom.

Flavienne Bruyant[1,2], Rémi Amiraux[1,2,3], Marie-Pier Amyot[1], Philippe Archambault[1,2], Lise Artigue[4], Lucas Barbedo de Freitas[2,5], Guislain Bécu[1,2], Simon Bélanger[2,5], Pascaline Bourgain[6], Annick Bricaud[7], Etienne Brouard[1], Camille Brunet[8], Tonya Burgers[9], Danielle Caleb[10], Katrine Chalut[11], Hervé Claustre[7], Véronique Cornet-Barthaux[8], Pierre Coupel[8], Marine Cusa[12], Fanny Cusset[1], Laeticia Dadaglio[13], Marty Davelaar[10], Gabrièle Deslongchamps[1,2], Céline Dimier[7], Julie Dinasquet[13], Dany Dumont[2,14], Brent Else[15], Igor Eulaers[16], Joannie Ferland[1,2], Gabrielle Filteau[1,2], Marie-Hélène Forget[1], Jérome Fort[17], Louis Fortier[1,2,‡], Martí Galí[1,18], Morgane Gallinari[3], Svend-Erik Garbus[16], Nicole Garcia[8], Catherine Gérikas Ribeiro[19,20], Coline Gombault[21], Priscilla Gourvil[22], Clemence Goyens[23], Cindy Grant[1,2], Pierre-Luc Grondin[1,2], Pascal Guillot[2,21], Sandrine Hillion[3], Rachel Hussherr[2], Fabien Joux[13], Hannah Joy-Warren[24], Gabriel Joyal[1,2], David Kieber[25], Augustin Lafond[8], José Lagunas[1,2], Patrick Lajeunesse[1], Catherine Lalande[1,2], Jade Larivière[1,2], Florence Le Gall[19], Karine Leblanc[8], Mathieu Leblanc[1,2], Justine Legras[3], Keith Lévesque[21], Kate-M. Lewis[24], Edouard Leymarie[7], Aude Leynaert[3], Thomas Linkowski[21], Martine Lizotte[1,2], Adriana Lopes dos Santos[26], Claudie Marec[1,27], Dominique Marie[19], Guillaume Massé[1], Philippe Massicotte[1,2], Atsushi Matsuoka[1,28], Lisa A. Miller[10], Sharif Mirshak[29], Nathalie Morata[3,12], Brivaela Moriceau[3], Philippe-Israël Morin[1,2], Simon Morisset[21], Anders Mosbech[16], Alfonso Mucci[30], Gabrielle Nadaï[1,2], Christian Nozais[11], Ingrid Obernosterer[13], Thimoté Paire[1], Christos Panagiotopoulos[8], Marie Parenteau[1,2], Noémie Pelletier[11], Marc Picheral[7], Bernard Quéguiner[8], Patrick Raimbault[8], Joséphine Ras[7], Eric Rehm[1,2], Llúcia Ribot Lacosta[1,31], Jean-François Rontani[8], Blanche Saint-Béat[1,32], Julie Sansoulet[1], Noé Sardet[29], Catherine Schmechtig[33], Antoine Sciandra[7], Richard Sempéré[8], Caroline Sévigny[2,14], Jordan Toullec[3], Margot Tragin[19], Jean-Éric Tremblay[1,2], Annie-Pier Trottier[1], Daniel Vaulot[19], Anda Vladoiu[34,35], Lei Xue[25], Gustavo Yunda-Guarin[1,2] and Marcel Babin[1].

[1] Takuvik international research laboratory (IRL3376), Université Laval (France)/CNRS (France), Québec G1V 0A6, QC, Canada

[2] Québec-Océan, Université Laval, Québec G1V 0A6, QC, Canada

[3] LEMAR, Univ Brest, CNRS, IRD, Ifremer, 29280 Plouzane, France

[4] LEGOS, University of Toulouse, CNRS, CNES, IRD, UPS, 31400 Toulouse, France

[5] Département de Biologie, Chimie et Géographie (groupes BORÉAS et Québec-Océan), Université du Québec à Rimouski, 300 allée des Ursulines, Rimouski G5L 3A1, QC, Canada

[6] Société AVUNGA, Lars en Vercors, France

[7] Laboratoire d'Océanographie de Villefranche, UMR7093, CNRS/Sorbonne Université, Villefranche-sur-Mer, France

[8] Mediterranean Institute of Oceanography (MIO), Aix-Marseille Université, Université de Toulon, CNRS, IRD, MIO, Marseille, France

[9] Centre for Earth Observation Science, University of Manitoba, Winnipeg MB, Canada

[10] Institute of Ocean Sciences, Fisheries and Oceans Canada, Sidney BC, Canada.

[11] Québec-Océan, Département de biologie, chimie et géographie, Université du Québec à Rimouski QC, Canada

[12] Akvaplan-niva, Fram Centre for Climate and the Environment, Tromsø, Norway

[13] Laboratoire d'Océanographie Microbienne (LOMIC), UMR7621, CNRS/ Sorbonne Université, Observatoire Océanologique de Banyuls-sur-mer, France

[14] Institut des sciences de la mer de Rimouski, Université du Québec à Rimouski, QC, Canada

[15] Department of Geography, University of Calgary, Calgary T2N 1N4, AB, Canada

[16] Department of Biosciences – Arctic Environment, Aarhus University, Denmark

[17] Littoral Environnement et Sociétés, UMR7266, CNRS/Université de La Rochelle, France

[18] Barcelona Supercomputing Center (BSC), Barcelona, Spain.

[19] ECOMAP, UMR7144, CNRS/Sorbonne Université, Station Biologique de Roscoff, France

[20] GEMA Center for Genomics, Ecology & Environment, Faculty of Sciences, Universidad Mayor, Santiago, Chile

[21] Amundsen Science, Université Laval, Québec QC, Canada

[22] Roscoff Culture Collection, FR2424 CNRS/Université Sorbonne, Station Biologique, Roscoff France.

[23] Operational Directorate Natural Environment, Royal Belgian Institute of Natural Sciences (RBINS), 29 Rue Vautierstraat, 1000 Brussels, Belgium

[24] Department of Earth System Science, Stanford University, Stanford, CA 94305, USA

[25] Department of Chemistry, College of environmental sciences and forestry, State University of New York, Syracuse, NY 13210, USA

[26] Asian School of the Environment, Nanyang Technological University, 50 Nanyang Avenue, Singapore 639798, Singapore

[27] Institut Universitaire Européen de la Mer, UMS3113, CNRS/Univ. Brest, Plouzane, France

[28] School of Marine Science and Ocean Engineering (SMSOE), Institute for the Study of Earth, Oceans, and Space (EOS) University of New Hampshire, Durham, USA

[29] Société Parafilm, Montréal QC, Canada

[30] GEOTOP and Department of Earth and Planetary Sciences, McGill University, Montréal QC, Canada

[31] Balearic Islands Coastal Observing and Forecasting System, SOCIB, 07122, Edificio Naorte, Bloque A, Parc Bit, Palma de Mallorca, Spain

[32] Dyneco Pelagos, IFREMER, BP70, 29280 Plouzané, France.

[33] OSU Ecce-Terra, UMS3455, CNRS/Sorbonne Université, PARIS Cedex 5, France

[34] LOCEAN-IPSL, UMR7159, CNRS/IRD/MNHN/Sorbonne Université, 75005 Paris, France

[35] Applied Physics Laboratory, University of Washington, Seattle, WA 98105, USA

‡ Deceased.

*Correspondence to:* Flavienne Bruyant (flavienne.bruyant@takuvik.ulaval.ca)

**Abstract.** The Green Edge project was designed to investigate the onset, life, and fate of a phytoplankton spring bloom (PSB) in the Arctic Ocean. The lengthening of the ice-free period and the warming of seawater, amongst other factors, have induced major changes in Arctic Ocean biology over the last decades. Because the PSB is at the base of the Arctic Ocean food chain, it is crucial to understand how changes in the Arctic environment will affect it. Green Edge was a large multidisciplinary collaborative project bringing researchers and technicians from 28 different institutions in seven countries together, aiming at understanding these changes and their impacts into the future. The fieldwork for the Green Edge project took place over two years (2015 and 2016) and was carried out from both an ice-camp and a research vessel in Baffin Bay, in the Canadian Arctic. This paper describes the sampling strategy and the dataset obtained from the research cruise, which took place aboard the Canadian Coast Guard Ship (CCGS) *Amundsen* in late spring/early summer 2016. The sampling strategy was designed around the repetitive perpendicular crossing of the marginal ice zone (MIZ), using not only ship-based station discrete sampling, but also high-resolution measurements from autonomous platforms (Gliders, BGC-Argo floats…) and under-way monitoring systems. The dataset is available at https://doi.org/10.17882/86417 (Bruyant et al., 2022).

## 1 Introduction

The Arctic Ocean is currently experiencing unprecedented environmental changes. The increase of the summer ice-retreat lengthens the phytoplankton growing season, but also increases the area of the marginal ice zone (MIZ). If trends are maintained, all Arctic sea-ice may become seasonal as early as twenty years from now (Meredith et al., 2019), increasing the MIZ coverage even more. Ice edge blooms represent much of the annual phytoplankton primary production in the Arctic Ocean (Perrette et al., 2011; Ardyna et al., 2013), and their current phenology is relatively well known (Wassmann and Reigstad, 2011; Leu et al., 2015). However, we currently do not know how precisely primary production will respond to climate changes. The overarching goal of Green Edge was to understand the processes that control an arctic PSB as it expands northward, and to determine its fate through the investigation of related carbon fluxes (e.g., Trudnowska et al., 2021). This study was also motivated by the discovery that PSBs can and do occur underneath the ice (Arrigo et al., 2014) despite the limited amounts of under-ice available light (Mundy et al., 2009; Arrigo et al., 2014; Lowry et al., 2014; Assmy et al., 2017, Randelhoff et al., 2019). Field studies for the Green Edge project were carried out in 2015 and 2016 at an ice-camp located on landfast sea ice close to Qikiqtarjuaq (NU, Canada). Additionally, during late spring/early summer 2016, a cruise aboard CCGS *Amundsen* was conducted in Baffin Bay. As explained in Randelhoff et al. (2019), Baffin Bay is both relatively easy to access and represents an ideal framework for this study, because environmental conditions are representative of what is observed at the pan-arctic scale. Particularly, the warm Greenland current flowing north on the Greenland side, and colder waters on the Canadian side flowing south (Baffin Island current) induce an evenly retreating ice edge, allowing a straightforward sampling strategy. From the Green Edge project, two separated data papers were produced. One paper by Massicotte et al. (2020) describes the dataset generated during two ice-camp campaigns on landfast ice (2015 and 2016). The latter paper is related to a dataset published on SEANOE: https://www.seanoe.org/data/00487/59892/, in 2019. The second paper (the present paper by Bruyant et al.) provides an overview of the dataset gathered during an oceanographic cruise conducted in Central Basin Bay in 2016. For more clarity, we created a second dataset on SEANOE associated with the present paper: https://doi.org/10.17882/86417. This 2022 DOI contains the final version of all the cruise data that could be formatted to be published by SEANOE (see section 6. Data availability). Note that the first dataset (Massicotte et al. 2020) contained some of the cruise data for the sake of contextualizing the ice-camp dataset.

## 2 Study area, sampling strategy and ship-based operations

For logistical reasons, the cruise was divided into two legs. Leg 1A started on June 3[rd] in Québec City, ended on June 23[rd] in Qikiqtarjuaq (NU), and included one week of transit to the study zone. Leg 1B started on June 23[rd] in Qikiqtarjuaq and ended in Iqaluit (NU) on July 14[th]. During the five-week period spent in Baffin Bay, the ship crossed the MIZ, from open waters (in the east) to sea ice-covered areas (in the west) and back again following latitudinal transects. A total of seven transects were covered between 68.0° N and 70.5° N (Fig. 1A). Three transects were covered during Leg 1A (68.5° N to 69.0° N) and four during Leg 1B (68.0° N and 69.5° N to 70.5° N). For each transect,

stations were separated by six nautical miles (approximately 11 km) to obtain a relatively high spatial resolution. Fifteen to twenty-five stations were sampled within each transect, for a total of 144 stations visited during the campaign.

The activities conducted at each type of station are detailed in Table 1. Briefly, at so-called CTD (Current Temperature Depth) stations, rosette casts did not include seawater collection. The rosette, a Sea-Bird model 32 carousel equipped with twenty-two 12-L Niskin bottles, was geared with multiple sensors (see Table 2 for details). At NUT stations, seawater samples were additionally collected at several depths between 2 and 2000 m for nutrient analyses. At BASIC stations, the apparent optical properties of seawater were measured using underwater profiling optical instruments. The variables measured at BASIC stations included the concentration of chlorophyll *a*, phytoplankton pigment, particulate carbon and nitrogen, and particulate absorption spectra. Finally, at FULL stations, a suite of measurements was made on seawater samples, and several underwater instruments were deployed as well, including a suite of optical sensors. Vertical plankton nets geared with a plankton imager, horizontal net trawls, benthic trawls and Ursnel box corers were also deployed at each FULL station. Note that the number of variables measured on seawater samples was significantly larger at FULL stations, where three rosette casts were necessary to cover the demand in water volume, compared with one rosette at BASIC stations. At least one FULL station was sampled in each of the three major domains covered by each transect, namely open waters, the MIZ and the ice-covered area. During Leg 1B a larger emphasis was placed on collecting ice-cores for analyses and measurements of light propagation through the ice and snow. FULL stations sampled in the ice-covered domain did not include trawling operations.

Our sampling strategy allowed successive crossings of different PSB stages: the early-bloom stage at the western end of transects covered in sea-ice, late- to post-bloom stages at the eastern end of transects in open waters, and full-bloom stage in the middle of transects around the ice edge (see Randelhoff et al., 2019). Between the stations, the ship-track water monitoring system (TSG SBE45 from Sea-Bird, WETStar fluorometer from WetLabs, LiCOR non-dispersive infra-red spectrometer (model Li-7000), Campbell Scientific CR1000 data logger) recorded temperature, salinity, chlorophyll-*a* fluorescence and $pCO_2$ at 7 m depth continuously when navigating outside the ice pack. A moving vessel profiler (MVP300-1700, AML Oceanographic, Victoria, BC Canada), equipped with a micro CTD (AML), a WetLabs C-Star transmissometer and a WetLabs Eco-FLRTD fluorometer; was deployed in open waters.

To significantly extend the monitoring of the PSB beyond the duration and space covered by the cruise, we deployed four BGC-Argo (Bio-Geo-Chemical-Argo) floats on July 9[th]. They collected data until late fall (Fig. 1B), performing a 0-1000 m profile each day. The ice-specific version of BGC-Argo floats that was deployed, the so-called PRO-ICE, is commercialized by NKE Electronics (France). These floats carry a typical biogeochemical payload (Sea-Bird 41 ARGO CTD, Aanderaa optode 4330 Oxygen sensor, Sea-Bird™ OCR-504 PAR (photosynthetic available radiation) and downwelling radiance sensor (380, 412 and 490 nm), Sea-Bird ECO-FLBBCD fluorescence chlorophyll-*a*, colored dissolved organic matter (CDOM), and backscattering sensor and Sea-Bird SUNA nitrate sensor). They also include a 2 way-directional Iridium communication Rudics for data transmission and a sea-ice detection system to protect the float from hitting sea ice on ascent (Le Traon et al., 2020; André et al., 2020). The 2016 data are available on the following website: http://www.obs-vlfr.fr/proof/ftpv/greenedge/db/DATA/FLOATS/.

Two SLOCUM G2 gliders (Teledyne Marine Inc.) were deployed during the cruise from the ship's zodiac either to revisit transects, or in areas that were not visited by the CCGS *Amundsen* (Fig. 1B). They both carried a similar scientific payload and communication system as the BGC-Argo floats, rendering the possibility of inter-calibration (Fig. 2). Gliders were primarily used in the MIZ where a 90-m icebreaker would disturb the fragile hydrological structure of shallow under-ice water masses. Gliders were deliberately directed through the same area as the BGC-Argo floats to compare CTD and optical data from both platforms. Results in Fig. 2 show a good agreement for the data. Gliders travel following a programmed sawtooth pattern, joining pre-defined waypoints. Data and instructions were transmitted both ways via iridium when the glider surfaced. Figure 3 shows an example of the level of detail provided by the gliders when travelling through complex water structures. Representing chlorophyll fluorescence measured over a three-week journey covering almost 500 km, the data show that the glider(s) did travel through both surface blooms and a subsurface chlorophyll maximum between 20-50 m. The data presented in Fig. 3 represents 693,000 chlorophyll fluorescence measurements. These constitute robust results towards the validation of the use of multiple measurement platforms to investigate complex systems.

A schematic synopsis of all operations carried out during the campaign can be found in Fig. 4A. The use of multiple analysis platforms allowed the measurement of more than 150 parameters during the Green Edge cruise (Table 3), together they accurately describe the complexity of the MIZ trophic systems (Fig. 4B) and most of the stocks and processes involved in the development of the PSB.

**3 Time-for-space formatting and data quality control**

One of the challenging tasks, when assembling data from a large group of researchers, is to adopt a common frame for spatial and temporal tagging of samples. Geographic positions require a lot of attention and conversion of latitude and longitude into one format (we used decimal degrees east) to ensure data could be easily merged. The concomitant use of local time and Coordinated Universal Time (UTC) during the cruise also presented a challenge. For the Green Edge cruise, an operation logbook was created to keep track of all operations conducted on the ship in sequential order during each day (local time). Each operation was associated with a unique operation ID to which all other data could be referenced. The use of ordinal date (number of days since January first) was used to avoid confusion between European and American date writing conventions. Each cast within a given operation type (CTD/RO for Rosette, AOP for Apparent Optical Properties, etc.) was numbered sequentially, starting from 001, throughout the entire cruise. As a result, any given operation received a unique code which thereafter could be used to merge all the data acquired during that operation.

Different control procedures were adopted to ensure the quality of the data. First, the raw data were screened to identify and –where possible, eliminate errors originating from the measurement devices, including sensor (systematic or random) errors inherent to measurement procedures and methods. Instrumental pre- and post-calibration corrections were applied when necessary. Statistical summaries such as average, standard deviation and range were computed to detect and remove anomalous values in the data. Then, data were checked for duplicates and remaining outliers. Once raw measurements were cleaned, data were structured and gathered into single comma-separated values (CSV) files.

Each of these files was constructed to gather variables of the same nature (e.g., nutrients). In each file, a minimum number of variables (columns) was always included to make dataset merging easier and accurate (Table 4).

**4 Description of data collection: an overview**

**4.1 Meteorological, navigational and ice coverage data**

Along the ship track, an Automated Voluntary Observing Ship (AVOS) system recorded all data related to navigation, including the position of the vessel and basic atmospheric meteorological data (including barometric pressure). A meteorological tower was also installed on the foredeck of the ship to measure additional variables (instantaneous wind speed and direction, air temperature and relative humidity, atmospheric partial pressure, and vertical fluxes of $CO_2$). Averaged wind speeds and directions over the entire Baffin Bay were retrieved from the Remote Sensing System website (http://www.remss.com/measurements/ccmp/) and were computed according to the Cross-Calibrated Multi-Platform (CCMP) wind vector analysis product (V2.0, Atlas et al., 2011) (see one example of a pattern calculated over the months of June and July in Fig. 5). A major change in wind patterns happened between June (light 1-2 m s$^{-1}$ southward winds) and July (4-5 m s$^{-1}$ northward winds), which impacted sea ice movements. Figure 6 shows sea-ice cover over four periods covering the total sampling time of the cruise (https://nsidc.org/data/g02186), NSIDC (U.S. National Ice Center and National Snow and Ice Data Center) (2010). The north-south general orientation of the ice edge is visible, along with, over time, the westward progression of the MIZ.

Ice cover history was compiled and expressed as Open Water Days (OWD) before sampling day (Fig. 7), calculated from the difference (in day number) between the date of sampling and the date at which the sea-ice concentration reached 10 (panel A), 50 (panel B) and 80% (panel C) in the geographical location under study. Ice concentration data was obtained from the Advanced Microwave Scanning Radiometer 2 (AMSR2) sea-ice concentration data on the 3.125 km grid (Spreen et al., 2008), downloaded from http://www.iup.uni-bremen.de:8084/amsr2data/asi_daygrid_swath/n3125/ (see Randelhoff et al., 2019, for details on the calculations). In Fig. 7, yellow and pale green colors on the Greenland side indicate a positive value between 20 and 40 OWD which reflects how long the open-water conditions had prevailed at those stations at the time of sampling. To the east, closer to the Canadian side, the colder colors and negative values indicate the presence of sea ice. Open water days is a useful metric/index and can be computed using different sea-ice cover (SIC) thresholds, depending on the goal. For example, in Randelhoff et al. (2019), the SIC value used for hydrological interpretation was 10% (as in Fig. 7 top panel). However, a biologist might want to consider a SIC of 80% (Fig.7 bottom panel) when looking for the onset of phytoplankton growth, as only 20% of open water surface already dramatically increases the amount of light available.

**4.2 Physical data**

**4.2.1 Sea ice**

During sea-ice sampling operations, snow depth, ice thickness and freeboard were measured at each ice coring site. Over the cruise, ice thickness varied between 32 and 108 cm, while freeboard varied between 10 and -8 cm (top of the

225 ice under water). Several cores were retrieved at each site using a 9 cm diameter Mark II ice-corer (Kovacs Enterprises Inc., Roseburg, OR USA). Each ice core was sliced into 10 cm segments (from the bottom) after temperature was measured. Salinity was assessed after thawing and filtration using a salinometer (Guildline Autosal 8400B, Guildline Instruments Ltd., Smith Falls, ON Canada). Snow density and granulometry were assessed opportunistically (Eicken et al., 2009).

### 4.2.2 Water masses

Hydrological conditions during the cruise were determined using several tools. A moving vessel profiler (MVP) was deployed, oscillating between 0 and 300 m depth while towed at an average speed of 12 knots, rendering a very high spatial resolution (Fig. 8 bottom row, one profile every 2 km). Data obtained with the MVP matched the patterns observed from the rosette data acquired on sampling stations (Fig. 8 middle row). Profiles of conductivity, temperature

and pressure were collected using a Sea-Bird SBE 911plus CTD system rigged on the rosette. The data were post-processed according to the standard procedures recommended by the manufacturer and averaged over 0.2 m vertical bins. While there was a sharp transition in SIC at the ice edge along transect 300, the change in SIC was less steep in the more extensive MIZ of transect 500 (top row in Fig. 8). Nonetheless, both transects show similar patterns, with 100% SIC, colder (below -1 °C) and fresher (salinity below 33.5 g kg$^{-1}$) waters close to the surface on the western

side, and 0% SIC, saltier (above 33.6 g kg$^{-1}$) and warmer (above 0 °C) waters within the first 50 m on the eastern side. These observations were consistent with the northward inflow of Atlantic-origin waters along the Greenland shelf break, and the southward outflow of Arctic/Pacific-origin waters along the Baffin Island shelf break.

Currents in the water column were measured using a hull-mounted 150 kHz Acoustic doppler current profiler (ADCP, Teledyne RD Instruments Ocean Surveyor, California, USA), as well as two L-ADCP installed on the rosette structure

(RDI, WHM300-1-UG304) in a Master/Slave configuration. Vertical profiles of water turbulence were measured at each FULL station using a self-contained autonomous microprofiler (SCAMP, Precision Measurement Engineering, California, USA) deployed from the zodiac. A detailed description of the hydrological structures in the studied area during the Green Edge cruise is presented in Randelhoff et al. (2019).

### 4.3 Chemistry

Partial pressure of $CO_2$ (p$CO_2$) was measured continuously (every two minutes) using a Li-7000 $CO_2$ analyzer (LICOR, Lincoln NE, USA) coupled to a General Oceanics Underway System model 8050 (General Oceanics, Miami FL, USA) connected to the ship-track water monitoring system. At each FULL, BASIC and NUT station, discrete samples were collected using the Niskin bottles at 10 or more depths for seawater analysis (see Table 3 for the complete list). To complement the p$CO_2$ data from the underway system and provide full profiles of the seawater $CO_2$ system,

total alkalinity and dissolved inorganic carbon (DIC) concentrations were determined on discrete samples according to Dickson et al. (2007). Concentrations of the major macronutrients (nitrate, phosphate and orthosilicic acid) were determined with a segmented flow AutoAnalyzer model 3 (Seal Analytical, Germany) using standard colorimetric methods adapted from Grasshoff et al. (1999). Nitrate concentration in the water column was also determined during

each CTD cast using an in situ ultraviolet spectrophotometer (ISUS, Satlantic Inc., Halifax NS, Canada) mounted on the rosette. Concentrations varied between 0 and non-limiting concentrations over the entire cruise, with concentrations gradually increasing from the surface to bottom. Surface waters showed higher concentrations of macronutrients in the western half of the transects than in the eastern half of the transects, indicating surface water nutrients had been used by the developing PSB.

Organic elemental composition of total and dissolved matter (nitrogen, carbon, and phosphorus) was measured on water samples taken from the Niskin bottles. Samples were immediately poisoned with sulfuric acid and brought back to the lab for analysis using wet oxidation, as described by Raimbault et al. (1999). Subtracting signals obtained for filtered samples (dissolved matter) from non-filtered samples (total matter) rendered calculated values for particulate organic matter. Particulate organic carbon and nitrogen were also analysed on filtered samples (Whatman™ glass fiber GF/F, GE Healthcare USA) using high-temperature oxidation combined with gas chromatography.

**4.4 Light Field and Bio-optics**

The characteristics of the light field (quantity and quality) passing through snow and sea ice into the water column were assessed, as light is the most important parameter triggering the PSB. From the top of the CCGS *Amundsen* wheelhouse, the total solar downwelling radiation was measured using a pyranometer (0.3 to 300 µm wavelength) and a radiometer (visible range 300 to 750 nm). Surface radiometry was performed using a HyperSAS (Hyperspectral Surface Acquisition System, SeaBird Scientific USA) placed on the bow of the ship, including simultaneous measurements of hyperspectral above-water downward irradiance ($E_d(0^+, \lambda)$), and sky and surface radiance ($L_{sky}(0^+, \lambda)$ or $L_{tot}(0^+, \lambda)$). Data were recorded at each FULL and BASIC station in open waters. Above-water hyperspectral remote sensing reflectance ($R_{rs}(\lambda)$) was calculated from the radiometric quantities following the ocean optics protocols of Mueller et al. (2003) and Mobley (1999). In-water vertical profiles of downward irradiance ($E_d(z)$) and upward radiance ($L_u(z, \lambda)$ or irradiance ($E_u(z, \lambda)$) in the water column were measured using two different versions of the Compact Optical Profiling System (C-OPS, Biospherical Instruments Inc.) radiometer. In open waters, the free-fall version of the profiler was deployed from the ship (Antoine et al., 2013). The sea-ice version (ICE-Pro) was deployed during ice sampling through an auger hole carefully filled with fresh snow to avoid, as much as possible, disturbing the underwater light field. A reference sensor provided simultaneous measurements of downward irradiance in the air. All measurements were made at 19 different wavelengths between 320 and 875 nm.

A profiling optical package was deployed at 28 stations to measure the inherent optical properties (IOPs) of seawater. The measured properties (and sensors) included: fluorescence of chlorophyll-*a* and fluorescent dissolved organic matter (FDOM) (WetLabs, Eco Triplets), spectral total non-water absorption coefficients between 360 and 764 nm (Hobilabs, a-sphere), particle backscattering coefficient at six different wavelengths (Hobilabs, hydroscat-6, 394, 420, 470, 532, 620 and 700 nm) together with CTD data (Sea-Bird SBE 19plus attached to the package). In addition, discrete water samples were taken at each FULL and BASIC station to measure in the lab the CDOM absorption coefficient ($a_{CDOM}$) between 200 and 722 nm using an Ultrapath (World Precision Instruments), and the phytoplankton and non-algal particle absorption coefficients between 200 and 860 nm determined from the "inside sphere" filter-pad

technique (Rottgers and Gehnke, 2012; Stramski et al., 2015) using a spectrophotometer equipped with a 155-mm integrating sphere (Perkin Elmer Lambda 19). Note that $a_{CDOM}$, phytoplankton and non-algal particle absorption coefficients were also measured on the bottom slice of thawed ice cores. The data revealed that the minimum light amount required for net phytoplankton growth (0.415 mol m$^{-2}$ d$^{-1}$; Letelier et al., 2004) can be reached deeper under the ice than expected (Randelhoff et al., 2019). Further details of the light field measurements can be found in Massicotte et al. (2020).

Three optical profilers were also attached to the rosette carousel and rendered 203 profiles of CDOM fluorescence (FluoCDOM Wetlabs USA) and chlorophyll concentration (estimated from in situ fluorescence, Seapoint fluorometer, USA), as well as 87 profiles of light transmittance (Wetlabs C-Star transmissometer, USA). Data is shown in Fig. 8 for transects 300 and 500. For both transects, the highest values of chlorophyll-*a* concentration and of the attenuation coefficient (both parameters being strong proxies for phytoplankton biomass) were observed close to the surface in the MIZ, and deeper at around 50 m in ice-free waters, showing a progression in the PSB development starting close to the surface along the ice edge and growing into a subsurface chlorophyll maximum (SCM) where surface waters were depleted in nutrients. Concentration of CDOM showed its lowest concentrations at the surface of open waters where SIC dropped below 50%, and where the surface waters had been depleted of nutrients by phytoplankton growth.

**4.5 Biodiversity**

The Green Edge project also aimed to understand the related potential impacts of evolving environmental conditions on Arctic food-webs in the context of climate change. Hence, great care was taken to sample the entire size spectrum of particulate matter and living organisms (Fig. 9) from the tiniest viruses and bacteria to demersal fishes, seabirds, and marine mammals. A wide variety of sampling techniques and analyses, from visual observation to highly automated underwater imaging systems, allowed us to ensure that almost all the levels of the trophic network were examined.

**4.5.1 Viruses and bacteria**

Abundance of viruses and bacteria was determined on fresh and preserved (glutaraldehyde 4% final concentration) water samples taken from the rosette at each FULL and BASIC station (10 depths) using two different flow cytometers. On board, fresh samples were counted using an Accuri™ C6 and preserved samples were counted back in the lab using a FACSCanto (both machines from Becton Dickinson Biosciences, San Jose, CA, USA). Samples were processed according to Marie et al. (2001). Bacteria (and viruses) are ubiquitous in the oceans, and in Baffin Bay we measured bacterial abundances up to $2.9 \times 10^6$ cells ml$^{-1}$.

For bacterial diversity analysis, water samples were filtered sequentially onto 20 µm, 3 µm (both polycarbonate filters, Millipore) and 0.22 µm (Sterivex-GV, Millipore). The filters and Sterivex were stored at -80 °C with RNAlater (Qiagen) until analyzed. The DNA/RNA co-extraction was carried out using the AllPrep DNA/RNA kit (Qiagen). The V4-V5 hypervariable region of the 16S rRNA gene was amplified by PCR using primers 515F-Y and 926R covering

a broad spectrum of diversity, including Archaea and Bacteria (Parada et al., 2016). PCR, as well as sequencing settings and bioinformatic of sequence data, can be found in Dadaglio et al. (2018).

A comparison of the bacterial diversity as a function of geographic location and size fractions (free living bacteria,
bacteria attached to particles smaller than 20 μm and larger than 20 μm) was made at a relatively broad taxonomic level (Fig. 10). The samples from the different groups were mainly dominated by Bacteroidetes and Proteobacteria. In general, the proportion of Proteobacteria decreased from ice stations to open water stations featuring a more advanced stage of PSB, giving way to Bacteroidetes and more specifically Flavobacteriaceae.

### 4.5.2 Phytoplankton community

Flow cytometry was used (same protocols as for viruses and bacteria, Section 4.5.1) to count and differentiate the smallest cells (picophytoplankton, nanophytoplankton, cryptophytes and *Synechococcus*) according to their fluorescence and scattering properties at each FULL station. At the surface, picophytoplankton and nanophytoplankton cell concentrations could reach 60,000 (station 719 – transect 7 station 19) and 9,000 (station 515 –transect 5 station 15) cell ml$^{-1}$, respectively while cryptophytes were always below 360 cell ml$^{-1}$. *Synechococcus*
cyanobacteria were never observed.

Some samples were used to start phytoplankton cultures, which were taken back to the Roscoff laboratory for purification using flow cytometry sorting, serial dilution, and single-cell pipetting. Pure cultures were characterized by microscopy and 18S rRNA gene sequencing. Most cultures isolated during the cruise belonged to diatoms, especially to the genera *Attheya* and *Chaetoceros* (Gérikas Ribeiro et al., 2020). All cultures were deposited in the
Roscoff    Culture    Collection    and    are    available    for    distribution    (http://www.roscoff-culture-collection.org/strains/shortlists/cruises/green-edge).

To study the phytoplankton community composition, an Imaging FlowCytobot (IFCB, McLane Research Laboratories Inc., East Falmouth MA, USA) was used during Leg 1B. The IFCB is best used for the study and identification of cells between 1 and 150 μm. Fresh samples (5 ml) taken from the rosette at each FULL station (all depths) and some
BASIC and NUT stations (2 to 7 depths) were analyzed, as well as samples from the 2 bottom-most slices of ice cores once melted. The IFCB takes pictures at a resolution of around 3.4 pixels per μm. Image descriptors/features were extracted with Matlab®, using scripts developed by Heidi Sosik (Sosik and Olson, 2007). Taxonomic determination was achieved using Ecotaxa (Picheral et al., 2017; http://ecotaxa.obs-vlfr.fr). Random forest algorithms were used for automatic classification. Reference set and validation of predictions were both done manually. Examples of specimens
observed during the Green Edge campaigns can be found in Massicotte et al. (2020).

A total of 203 underwater vertical profiles were acquired using an underwater vision profiler (UVP, model 5-DEEP, Hydroptics France) installed on the frame of the rosette carousel. The UVP5 collects in-focus images in the small seawater volume lit by its light emitting diodes (LEDs), as it is lowered in the water column. An automated computer system (https://ecotaxa.obs-vlfr.fr/) was used to sub-sample images of individual objects and sort them into the
appropriate category (marine snow or various taxa of zooplankton). The UVP has been mainly developed to count and identify particles larger than 100 μm. Figure 11 (left panels) show the average vertical profiles of particle concentration

(ml$^{-1}$) over the top 350 m of the water column, for open-water (top) and under-ice (bottom) stations. The number of particles close to the surface in open waters coincides with the larger phytoplankton biomass observed there compared with ice-covered stations, consistent with lower primary and secondary production under sea ice. Deeper in the water column, around 300 m, the large particle concentrations observed at open-water stations likely reflect resuspension of bottom sediments, because these observations were mostly made in the eastern part of Baffin Bay over the continental shelf, whereas under-ice stations were mostly located on the deeper Canadian side of the Bay (see the bathymetry in Fig. 1).

Samples for taxonomic analyses of micro-algae by microscopy were taken at each FULL and BASIC station at ten sampling depths. Half a litre of seawater was preserved with Lugol and kept at 4 °C until it was analyzed in the laboratory. Visual observation and taxonomic determination were done using an inverted microscope (Eclipse TS100, Nikon Instrument Inc.) according to the Utermöhl method (Utermöhl, 1958), using 25 or 50 ml columns. Three transects of 26 mm at 400x were systematically observed for identification and counting of Bacillariophyceae, Dinophyceae, flagellates and ciliates. Larger phytoplankton cells and colonies were observed in all chambers at 100x. Diatoms (Bacillariophyceae) were found at every station, primarily at the surface of the water column, along with flagellates (both at the surface and in the subsurface chlorophyll maximum, SCM) (Fig. 12). The most striking feature was the dominating presence of a *Phaeocystis* sp. (Prymnesiophyceae, blue bars in Fig. 12) at the SCM, reaching 60 to 90% of the cell counts (and to a lesser extent at the surface) across a wide range of ice cover conditions (OWD values between -12 and 12 days).

Phytoplankton and ice-algae pigments were measured to derive indices of micro-algae biomass and taxonomic composition, and to get information on processes such as photoacclimation, senescence and grazing activities (Roy et al., 2011). Rosette water samples were filtered onto GF/F filters (Whatman™, GE Healthcare Life Sciences) and quickly frozen in liquid nitrogen. Back in the land-based laboratory, samples were thawed and extracted in 100% methanol, separated, and identified by HPLC, as described by Ras et al. (2008). Twenty-five individual pigments or groups of pigments were identified and quantified at each FULL and BASIC station (10 depths sampled each time). Figure 13 shows the distribution of total chlorophyll-*a* and pheophorbide concentrations along transects 300 and 500. The high chlorophyll-*a* concentrations close to the surface in the MIZ, and deepening towards open waters in the east, confirmed the evolution of the PSB from an under-ice bloom to a SCM. Note that highest concentrations of pheophorbide were systematically found underneath the accumulation of chlorophyll *a*, indicating the sinking of degrading phytoplanktonic material.

### 4.5.3 Zooplankton and Fish

Zooplankton represents the second level of the food chain. The UVP5 and the Imaging FlowCytobot (see previous sections) both rendered valuable information on small zooplankton specimens (below 150 µm). Figure 11-right panels show copepod volumetric fraction (cm$^3$ m$^{-3}$) over the top 350 m of the water column, for open-water (top panel) and under-ice (bottom panel) stations. The total copepod volumetric fraction calculated based on automatic identification made using the Ecotaxa web application (http://ecotaxa.obs-vlfr.fr), shows subsurface peaks around 20 m at both

open-water and ice-covered stations. A secondary peak right at the surface is present at ice-covered stations, where a subpopulation of copepods may stay close to the bottom of sea ice to feed on sympagic microalgae, small animals, and related detritus.

For bigger specimens, a series of vertical nets and trawls were deployed at each FULL station (no trawling operations took place when sea ice was present). An assembly of 4 nets of 3 different mesh sizes (50, 200 and 500 μm) coupled with a Lightframe Onsight Key species Investigation (LOKI) system rendered high-resolution pictures of individuals sampled along the water column together with actual specimens. The multi-net plankton sampler (Hydro-bios, Altenholz, Germany) uses a different sampling strategy. Composed of 9 identical nets (200 μm mesh size), it is hauled

vertically in the water column with the nets opening sequentially at different depths, each net collecting a slice of the water column displaying the vertical distribution of the species sampled. Figure 13C shows abundance (ind m$^{-3}$) of zooplankton sampled using the vertical 200 μm mesh net along transects 300 and 500. Copepods represent the main zooplankton class observed during the cruise, and they were present at every sampling site. The particularly high abundance at station 507 (situated under the ice where a phytoplankton bloom had already disappeared) might be due

to the relatively high abundance of copepod nauplii (25% of all copepod individuals compared to the usual 4% at other stations).

Ichthyoplankton was sampled using a double square net towed obliquely from the side of the ship at a speed of ca. 2-3 knots to a maximum depth of 90 m. A Star-Oddi® mini-CTD attached to the frame and flowmeters determined the real depth and volume of sampling. For fish sampling, an echo sounder (EK60, Simrad, Kongsberg Maritime,

Norway) mounted on the hull was used to locate and determine the depth of fish aggregations along the ship track during the entire cruise. When pelagic juveniles and adult fish were present at the sampling stations, an Isaac-Kidd Midwater Trawl (IKMT, Filmar and Québec-Océan, Québec, Canada) was towed for 20 minutes at a speed of 2 to 3 knots. When demersal fish were detected, a Benthic Beam Trawl (BBT, Filmar and Québec-Océan, Québec, Canada) was used to sample bigger specimens (between 10 and 32 mm mesh size) living on the bottom sediment.

Samples of zooplankton were all processed in the same way. Swimmers (fish larvae and juveniles) were sorted out, measured, identified, and preserved in a mix of 95% ethanol and 1% glycerol (final concentrations) for later analysis, while zooplankton samples were preserved in 4% formaldehyde solution. Zooplankton abundance and diversity were determined using binoculars back at the laboratory. A few samples (hydro-bios) were analyzed using the zooscan and the ecotaxa identification tools (https://ecotaxa.obs-vlfr.fr/prj/802). Fish from IKMT and BBT sampling were sorted,

counted, identified, and measured before preservation in a -20 °C freezer in case further analyses are needed in the future. Acoustic data from the EK60 were analyzed in Echoview® (see Geoffroy et al. (2016) for details).

### 4.5.4 Benthos sampling

Benthos sampling was performed to answer the following specific objectives to test if i) sea-ice cover is the primary environmental driver of contribution and geographic distribution of sympagic carbon on the seabed, ii) sympagic

carbon is the most important baseline food source supporting benthic consumers during spring and summer in areas close to the MIZ, and iii) deep benthic food web dynamics and structural variability are directly linked with both depth

and availability of food sources (Yunda-Guarin et al., 2020). The sampling was achieved using two different strategies; a total of 16 Agassiz trawling and 34 box coring operations (some down to more than 2000 m depth) were carried out during the cruise. The Agassiz trawl (KC Denmark a/s Research Equipment) is a medium-size dredge trawled behind the ship, allowing sampling of the macrofauna living on the sediment surface. Once brought back onboard, the content of the net was immediately rinsed with seawater, manually sorted and identified to the lowest taxonomic level possible. Samples were brought back to the laboratory to be identified under a dissecting microscope when onboard identification was impossible. Figure 14 shows an example of the diversity of benthic organisms. More than 220 species of macrofauna were identified during the Green Edge cruise from more than 25 classes (Grant C. and Yunda-Guarin G. Unpublished data). A box corer was used to sample sediment from each FULL station. The sediment samples were then divided among the research teams for diverse analyses (see Table 3 for a complete list) including (but not limited to) identification of organisms living inside the sediment, incubations for respiration and nutrient utilization or chemical analysis.

### 4.5.5 Birds and marine mammals

During the entire cruise, a systematic bird and marine mammals survey was carried out from the ship's wheelhouse (see LeBlanc et al., 2019 for detailed methodology). A total of 20 different bird species and 8 different mammal species were identified. Northern fulmar, thick-billed murre and little auk were the most common bird species observed. Ringed seal, hooded seal and harp seal were the most common seals. The long-finned pilot whale was the most common whale species observed. A total of 10 polar bears were observed.

A total of 123 seabirds from 7 species were also collected from a zodiac deployed from the CCGS *Amundsen* in Greenland waters between June 10[th] and July 8[th]. This includes black-legged kittiwakes (*n*=8), glaucous gulls (*n*=6), great black-backed gull (*n*=1), little auks (*n*=19), northern fulmars (*n*=42) and thick-billed murres (*n*=36). Sampled birds were frozen at -20 °C until laboratory analyses. A first study aimed to investigate the co-distribution of seabirds and their fish prey along the MIZ (LeBlanc et al., 2019). To this end, stomach contents were examined for 74 birds (35 murres, 30 fulmars and 9 kittiwakes) under a dissecting microscope. Otoliths were retrieved and used to identify fish species, age and size. A second focus was recording of plastic in the stomachs. Plastic data are used in OSPAR monitoring and AMAP working groups on plastic pollution. A third study aimed to determine bird's association to sea ice and ice-derived resources by the combination of different trophic markers. Hence, liver, muscle, and blood (from the cardial clot) samples were collected from a total of 52 bird carcasses (27 murres, 14 auks, 3 kittiwakes and 8 fulmars), on which Highly Branched Isoprenoids (HBIs), carbon and nitrogen stable isotopes and fatty acids were measured. Finally, a fourth study was looking at stable isotope data together with mercury (Hg) data for both muscle and liver.

## 5. Biological production and fluxes

### 5.1 Bacterial production, respiration, and viability

At each FULL station during the cruise, water samples were taken from 2-3 depths (surface, deep chlorophyll maximum (DCM) and below DCM) to determine bacterial respiration. Oxygen concentration was determined using the Winkler method on 1-µm filtered samples before and after a 5-day incubation in the dark at 1.5 °C. Bacterial respiration varied overall between 0 and 1.63 µmol $O_2$ $l^{-1}$ $d^{-1}$, with a mean value of $0.35 \pm 0.41$ µmol $O_2$ $l^{-1}$ $d^{-1}$. For bacterial production determination, water was collected at each FULL station from 8-10 depths. Bacterial production

was measured by [$^3$H]-Leucine incorporation (Kirchman et al., 1985) modified for microcentrifugation (Smith and Azam, 1992). Overall values varied between 0 and 1.51 µgC $l^{-1}$ $d^{-1}$ around a mean value of $0.17 \pm 0.25$ µgC $l^{-1}$ $d^{-1}$. The use of the propidium monoazide (PMA) method identified a high bacterial mortality in sea ice (up to 90%) and in SPM material (up to 68%) collected in shallower waters at station 409 and station 418 (Burot et al., 2021).

### 5.2. Primary production and micronutrient cycling

To determine the fate of the phytoplankton spring bloom, one must first determine primary production. In situ simulated incubations were carried out at each FULL station on water sampled from the rosette at 8 to 10 depths determined as chosen percentages of surface photosynthetic available radiation (PAR, namely 100, 50, 25, 10, 6, 2.9, 1.2, 0.6 and 0.1%). The melted bottom-most slices of ice cores were also incubated, when available. After spiking the water with a mix of $^{13}C/^{15}N$ tracers, samples were incubated on deck at simulated light levels identical to the sampling

light levels. The dissolved and particulate matter resulting from these incubations were analyzed by mass-spectrometry resulting in detailed nitrogen assimilation and regeneration values (see Table 3 for a complete list of measurements), as well as phytoplankton primary production (PP). Primary production varied between 0 and $88.13 \pm 3.0$ µgC $l^{-1}$ $day^{-1}$ over the entire cruise.

    Photosynthetic parameters also allow calculation of primary production and provide insight into the efficiency and

characteristics of photosynthesis of a given sample. Onboard, photosynthetic parameters were determined by the P *vs* E curves method using $NaH^{14}CO_3^-$ spiked incubations of water samples (Lewis and Smith, 1983). Changes in the saturation parameter $E_k$ (µmol Quanta $m^{-2}$ $s^{-1}$) in the surface waters of the transect show clear variations in light level acclimation, increasing from lower values around 39 µmol Quanta $m^{-2}s^{-1}$ at the western under-ice stations to 217 µmol Quanta $m^{-2}$ $s^{-1}$ at the eastern open-water stations (transect 700 values given as an example). Great emphasis was placed

on the contribution of diatoms to primary production, as diatoms are the main phytoplankton group present during the PSB. Experiments on silica production and dissolution were performed throughout the cruise to locate the actively growing diatoms. These experiments confirmed the occurrence of active silicification beneath the sea ice, where both centric and pennate diatoms were observed (see details in Lafond et al. (2019)).

### 5.3 Fate of the phytoplankton spring bloom

Some organic matter produced by the PSB was exported down the water column, as algal cells aggregated and sank, or were grazed upon by vertically migrating zooplankton. One ambitious experiment was conducted during the cruise

to monitor the export of the PSB at a high temporal resolution. A sequential sediment trap (Technicap PPS4 France; 12 sampling cups) was anchored to an ice floe and deployed 25 m under the ice from June 15th to July 9th, 2016. Sediment trap collection cups were filled with filtered seawater adjusted to a salinity of 38 psu with NaCl and a formalin concentration of 4% to preserve samples during deployment and after recovery. The carousel holding the sampling cups was programmed to rotate every 2 days. The sediment trap was deployed in the marginal ice zone along transect 200 (Fig. 1), eventually drifted south with the ice, and was recovered on the way back to Iqaluit. The sediment trap was no longer anchored to its floe at recovery, but sea ice was still present in the region. Taxonomic identification of the algal cells collected showed a constant export of diatoms ($\sim$50 million cells m$^{-2}$ d$^{-1}$) from June 15$^{th}$ to early July, when a 6-fold increase in diatom fluxes was observed from July 5$^{th}$ to July 7$^{th}$ ($\sim$300 million cells m$^{-2}$ d$^{-1}$) along with a peak in chlorophyll-*a* fluxes. More than half of the cells exported during the peak in algal fluxes were identified as the ice-associated pennate diatom *Navicula* spp.. Fluxes of the ice-obligate pennate diatom *Nitzschia frigida*, among the first species to be consistently exported from the melting sea ice in the Arctic Ocean (Lalande et al., 2019; Dezutter et al., 2021; Nadaï et al., 2021), peaked from June 23$^{rd}$ to June 25$^{th}$, probably indicating the onset of sea ice melt. Fluxes of copepod fecal pellets collected in the sediment trap were higher prior to June 27$^{th}$, suggesting under-ice grazing of ice algae until the ice melted.

### 5.4 Benthic processes

Taken from the box corer, portions of the sediment were incubated at in situ simulated conditions of temperature and light to assess the consumption of oxygen and nutrients by endofauna. Oxygen use in the sediment cores allowed calculation of the benthic carbon demand (mgC m$^{-2}$ d$^{-1}$) which was found to be especially high at stations in open waters where the PSB had already reached senescence and sinking organic matter had reached the bottom.

### 5.5 Other data

The exhaustive list of parameters measured during the cruise is presented in Table 3 along with the responsible principal investigators (PIs) name.

### 6. Data availability

Making such a large and diverse dataset available to others requires the use of many different platforms. Administrative rules and previous habits and commitments explain why our dataset is hosted by various websites with some of it in more than one place (see in Table 3, the link and file information, when applicable for each parameter acquired during the cruise). Some funding Agencies require data to be deposited in a specific database as a deliverable. In our case, the "Les Enveloppes Fluides et l'Environnement-Cycles Biogéochimiques Environnement et Ressources" (LEFE-CYBER) repository is our main host: http://www.obs-vlfr.fr/proof/php/GREENEDGE/x_datalist_1.php?xxop=greenedge&xxcamp=amundsen. This is where ALL data and associated metadata can be found for the Green Edge cruise. Particularly, detailed metadata files associated with each variablecontain the principal investigator's contact information. For specific questions, the PI associated with the data

should be contacted directly. The LEFE-CYBER platform, however, does not deliver DOI, which is a very important feature for visibility of data. To obtain a DOI for the Green Edge cruise dataset, we uploaded the available formatted data on SEANOE (SEA scieNtific Open data Edition) under the CC-BY license: https://www.seanoe.org/data/00752/86417. All data hosted on the SEANOE website have been formatted as described

in section 3, but not all data original format allows transformation into the ".csv" file type. It is therefore important to keep a repository up to date where one can find all raw data.

Major long-term research programs often have their own repository / database available which has been used since the onset of their research. While the data of the BGC-Argo floats we deployed during the Green Edge cruise are hosted on the LEFE-CYBER repository, they also have been made available, together with the entire BGC-Argo data,

from the biogeochemical Argo database: https://biogeochemical-argo.org/data-access.php

Some more specific data acquired during the cruise are also available on dedicated websites. For example, dissolved inorganic carbon (DIC), alkalinity and $^{18}$O data are also archived with the Ocean Carbon and acidification Data System (OCADS): https://doi.org/10.25921/719e-qr37.

Geographical specificity might also be a motivation for cross-uploading of data. Since the *Amundsen* has been used

to conduct polar research, all navigation, AVOS, ADCP, MVP and CTD data are systematically uploaded to the Polar Data Catalog (PDC): https://www.polardata.ca/.

Please note that in Table 3 only one address for each parameter is provided while most of them are also available from other sources.

## 7. Lessons learned

As for any scientific cruise, a large amount of data was acquired by many people. Even though guidelines had been suggested ahead of time for data formatting, merging and storage, a tremendous amount of effort was necessary to collect, assemble and standardize the data. It is important that a clear and streamlined data management plan be established ahead of time to avoid errors or loss of data in the merging process. For oceanographic CTD-rosette sampling-based cruises, depth of sampling necessitates special attention, as it is crucial to use Niskin bottle number

instead of nominal depth to correctly merge the data. Furthermore, we cannot emphasize enough that data management specialists must be involved from the beginning of such large-scale projects, to ensure that data is properly documented, to render the best quality dataset possible and avoid loss of both valuable time and data.

## 8. Conclusion

The Green Edge cruise was of typical oceanographic design. In terms of goal achievement, the cruise was extremely

successful and generated an impressive dataset over a diverse set of disciplines, providing a global picture of the explored environment and of all the processes fuelling the Arctic food-web. Figure 4B represents all interactions existing and/or measured during the cruise between compartments of the various trophic levels. The generated dataset contains a much larger number of parameters than those presented in this paper. All data can be obtained from the

data repository and provide an excellent opportunity for re-use and comparison with other Arctic datasets. A special issue of the Elementa: Science of the Anthropocene journal entitled "Green Edge –The phytoplankton spring bloom in the Arctic Ocean: past, present and future response to climate variations, and impact on carbon fluxes and the marine food web" contains a collection of research papers referring to this cruise. A complete list of peer reviewed journal publications presenting data from either or both the Green Edge Ice Camp or Green Edge cruise can be found in Table 5.

## 9. Authors contribution

MB designed the Green Edge project, including the scientific objectives and sampling strategy. MB, KL, FB, TL, PG, GJ, TB, MP, CM, KC, DM, MT, GD, GF, LD, JD, CL, ML, GN, NM, MC, MP, K-ML, HJ-W, ER, AV, LBdF, DD, NG, HC, AS, BQ, SH, GB, SEG, CG, P-LG, JET, EB, CS, MGT, JR, AB, RA, CB, BM, JL, EL, PB and PC were onboard the ship and took part in sampling and on-board analysis. FB, PG, GJ, TB, MP, CM, KC, DM, MT, GD, LD, GF, LD, JD, CL, LM, GN, NM, NP, K-ML, HJ-W, ER, AV, LBdF, DD, NG, HC, AS, BG, SH, GB, GC, P-LG, M-NH, CS, MGT, JR, AB, RA, CB, BM, JL, EL, BS-B, PC, PA, LA, SB, DC, VC-B, FC, MD, CD, BE, IE, JF, LF, MG, CGR, CG, PG, CG, SH, RH, FJ, AL, PL, FLG, KL, JL, AL, ML, ALdS, GM, AM, LAM, P-IM, AM, AM, CP, MP, PR, J-FR, RS, JT, A-PT, DV and CN took part in processing and analyzing of the samples and in generating data. M-PA, and PM cleaned, merged, and assembled the dataset. CS maintains the Les Enveloppes Fluides et l'Environnement-Cycles Biogéochimiques Environnement et Ressources (LEFE-CYBER) repository it is stored in. NS, SM, JS, LRL, TP and PB oversaw communication and outreach. MHF, JF, JL and FB oversaw logistics. FB wrote the manuscript.

## 10. Acknowledgments

The Green Edge project was funded by the following French and Canadian programs and agencies: Agence nationale de recherche (ANR) (Contract #111112), ArcticNet, Canada Excellence Research Chair (CERC) on Remote sensing of Canada's new Arctic frontier, le Centre national d'études spatiales (CNES, project #131425), French Arctic Initiative, Fondation Total, the Canadian Space Agency, Fisheries and Ocean Canada, Sentinelle Nord, LEFE and Institut Paul-Emile Victor (IPEV, project #1164). This project was conducted using the Canadian research icebreaker CCGS Amundsen with the support of the Amundsen Science program funded by the Canada Foundation for Innovation (CFI) Major Science Initiatives (MSI) Fund. We wish to thank the officers and crew of the CCGS *Amundsen*. The project was conducted under the scientific coordination of the CERC on Remote sensing of Canada's new Arctic frontier and the Centre national de la recherche scientifique CNRS/Université Laval Takuvik Joint International Laboratory (IRL3376). The field campaign was successful thanks to the contributions of G. Bécu, J. Lagunas, D. Christiansen-Stowe, J. Sansoulet, E. Rehm, M. Benoît-Gagné, M.-H. Forget and F. Bruyant from Takuvik laboratory and J. Bourdon, C. Marec and M. Picheral from CNRS. We also thank Québec-Océan and the Polar Continental Shelf Program for their in-kind contribution in terms of polar logistics and scientific equipment. We thank

Marie-Pier Amyot for data cleaning and Étienne Ouellet for IT support and data infrastructure management. We thank E. Trudnowska and one anonymous referee for their numerous and useful comments.

The authors declare that they have no conflict of interest.

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

**FIGURES**

A

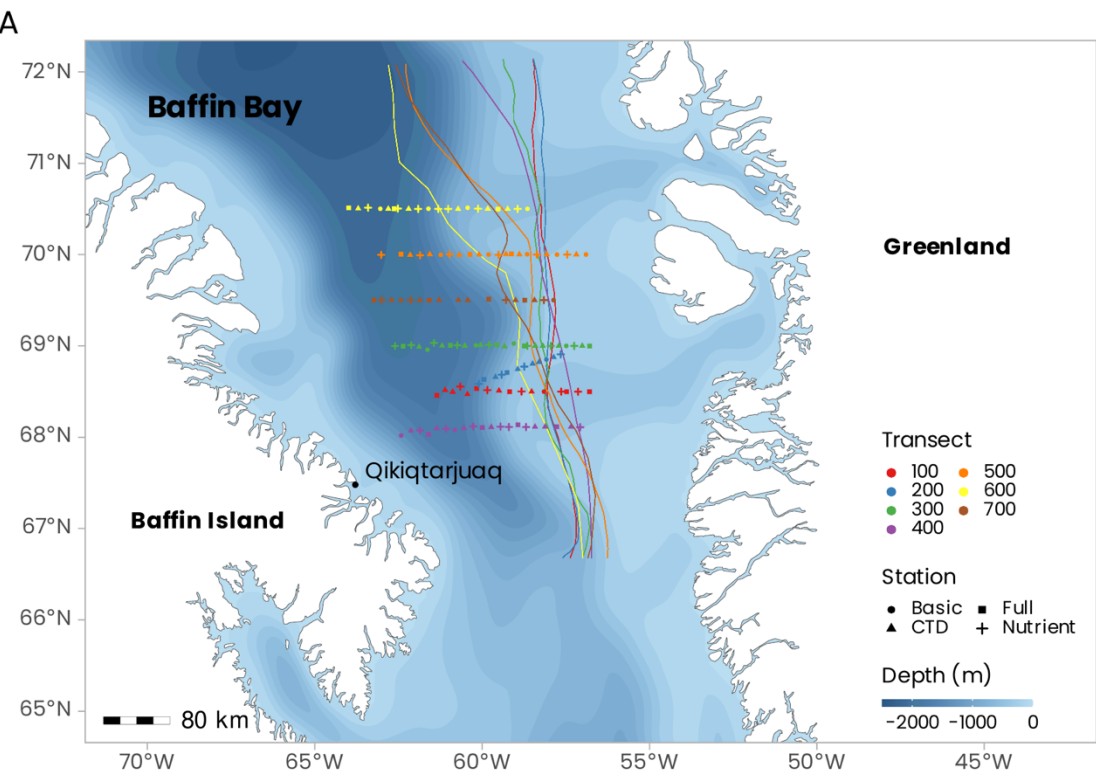

B

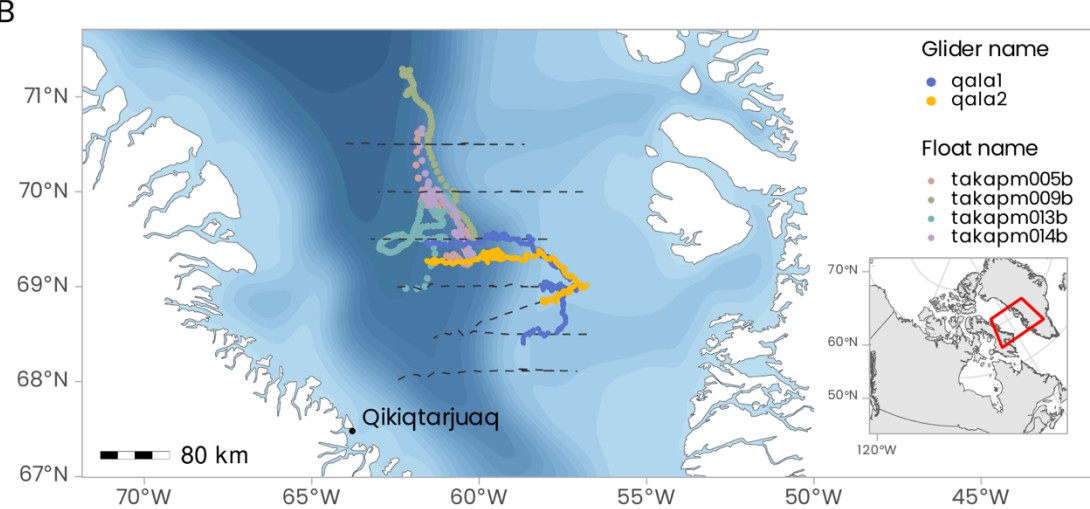

**Figure 1: Cruise maps showing A. the seven transects and all stations with shape distinction between station type (FULL, BASIC, NUT and CTD, see Table 1 for description). Colored lines indicate the position of the ice edge (sea ice concentration at 80%) on each transect at date of sampling; ice cover persisted on the western side of Baffin Bay, while the eastern side cleared earlier. Starting date of each transect sampling is as follows: June 9th (100), June 14th (200), June 17th (300), June 24th (400), June 29th (500), July 3rd (600) and July 7th (700). B. Tracks of the 4 BGC-Argo floats deployed during the Green Edge cruise over their first 101 days of life, and the journey of the two Slocum gliders (blue and orange). The dashed line represents the ship track along the seven transects.**

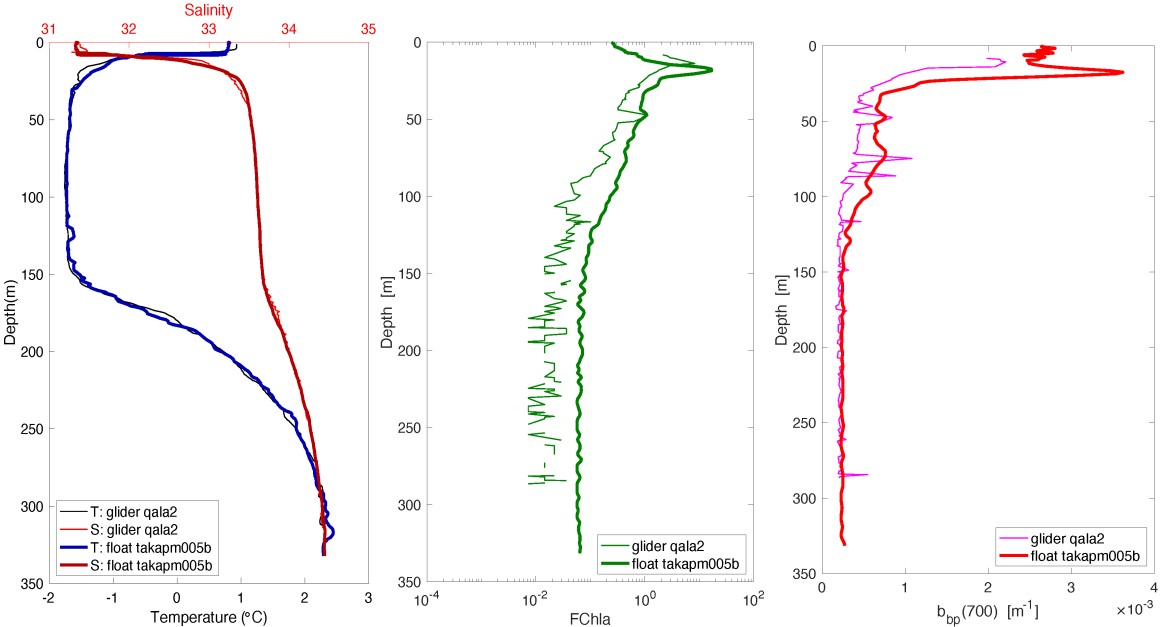

**Figure 2: Comparison between salinity (psu), temperature (°C), chlorophyll fluorescence (relative units) and backscattering at 700 nm (bbp(700), m$^{-1}$) data from the qala2 glider and the takapm005b BioArgo float at their closest common position (Fig. 1B).**

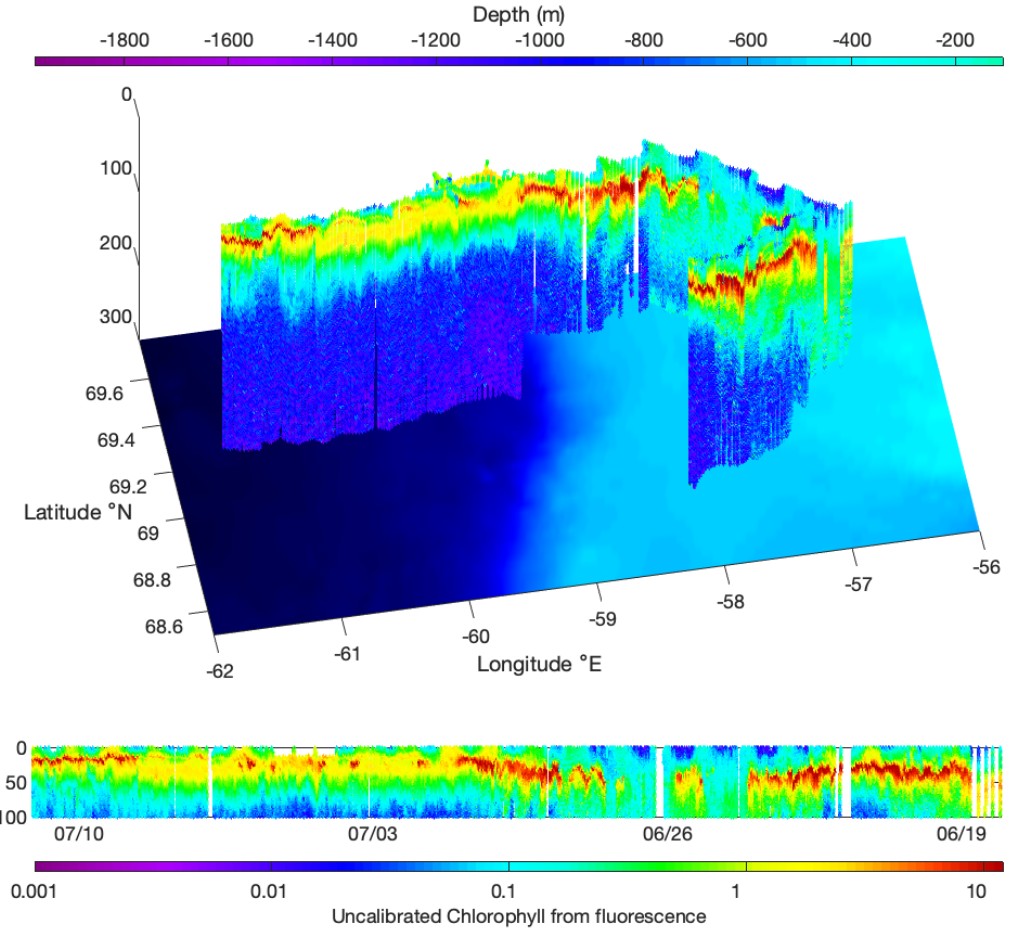

**Figure 3: Chlorophyll fluorescence measured by the Glider qala2 during its 23-day journey (Fig. 1B).**

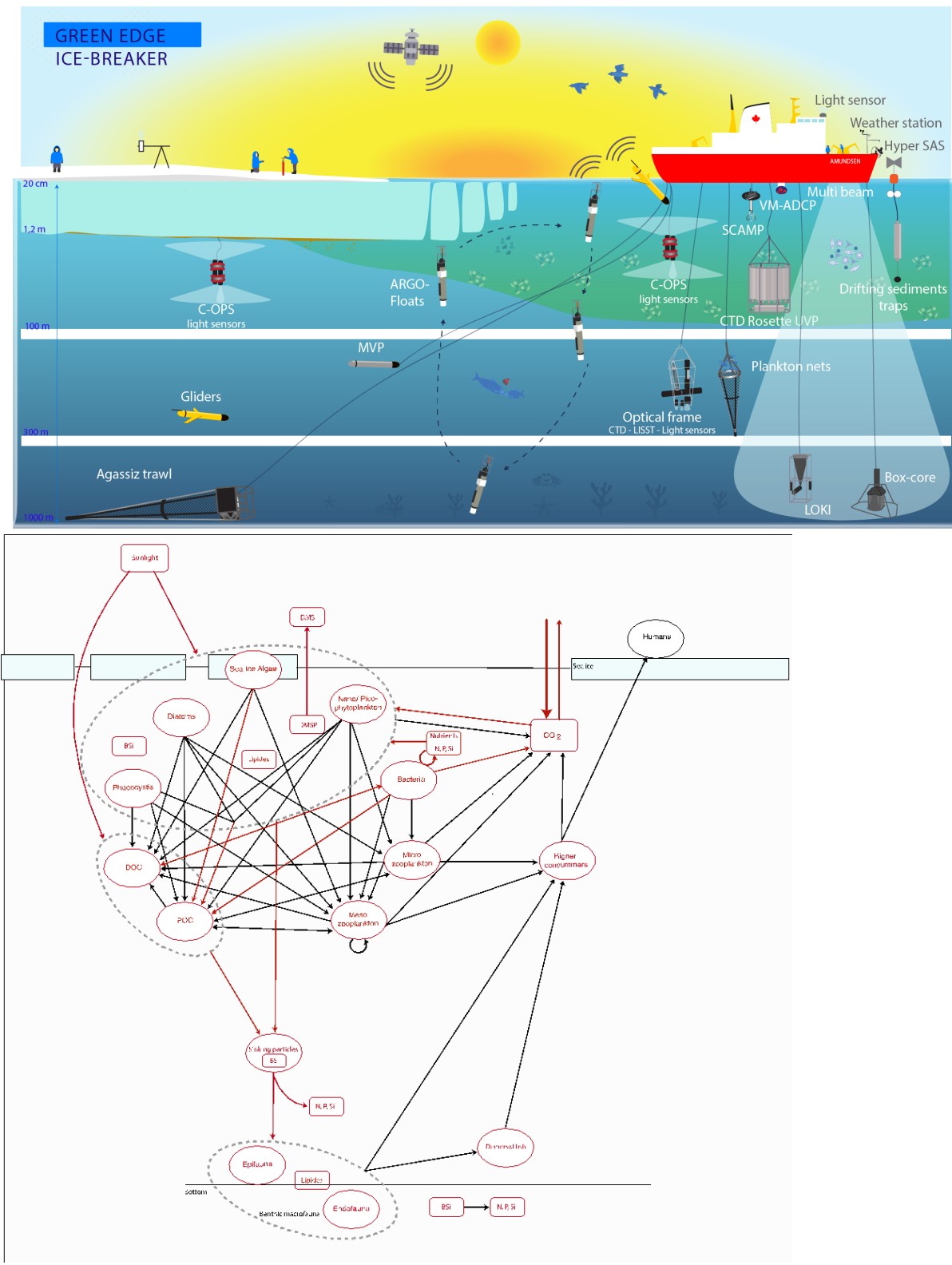

**Figure 4: Figure 4A: Representation of all operations carried out during the Green Edge campaign. B: Schematic representation of all stocks (bubbles) and fluxes (arrows) measured during the Green Edge cruise as part of the MIZ trophic system. Measured and calculated variables are represented in red.**

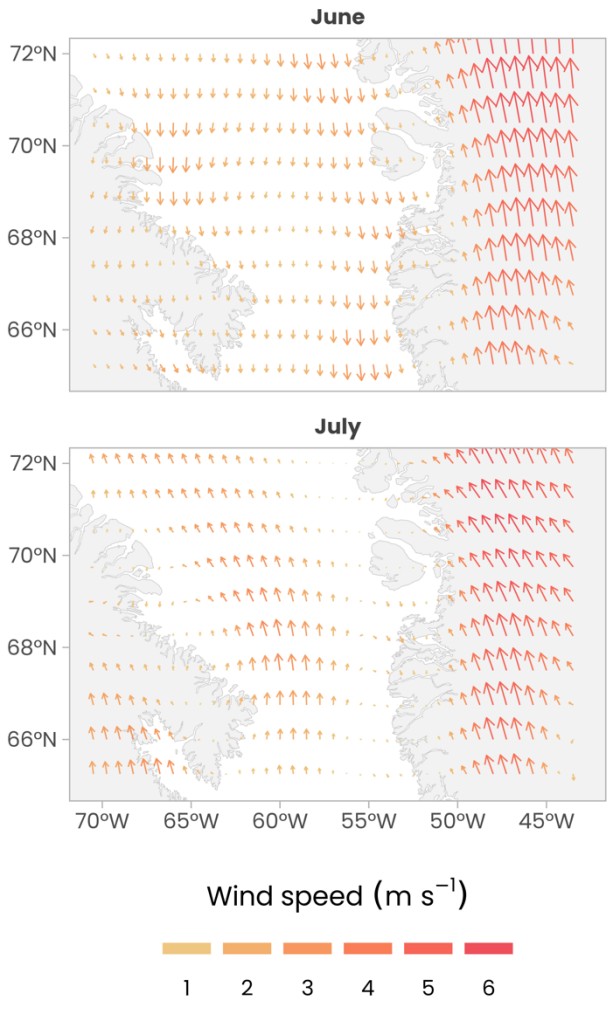

**Figure 5: Average wind direction and speed (arrow length and color) over Baffin Bay in June 2016 (top panel) and July 2016 (bottom panel); CCMP wind vector analysis product (V2.0, Atlas et al., (2011)).**

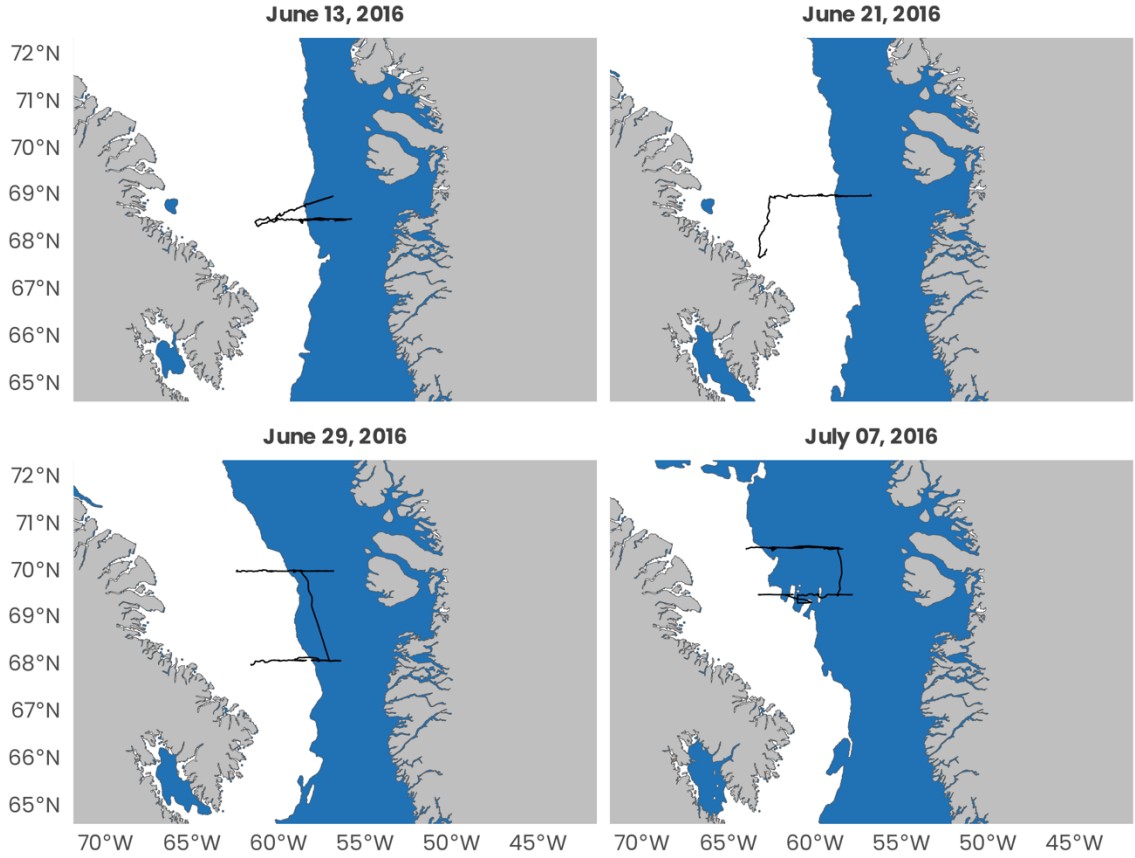

**Figure 6: Average weekly sea-ice extent for each of the four 8-day periods during the Green Edge cruise. Date is the median date of each 8-day sampling period. The white areas represent sea ice and the blue areas the open water. Black lines represent the ship route during said 8-day period.**

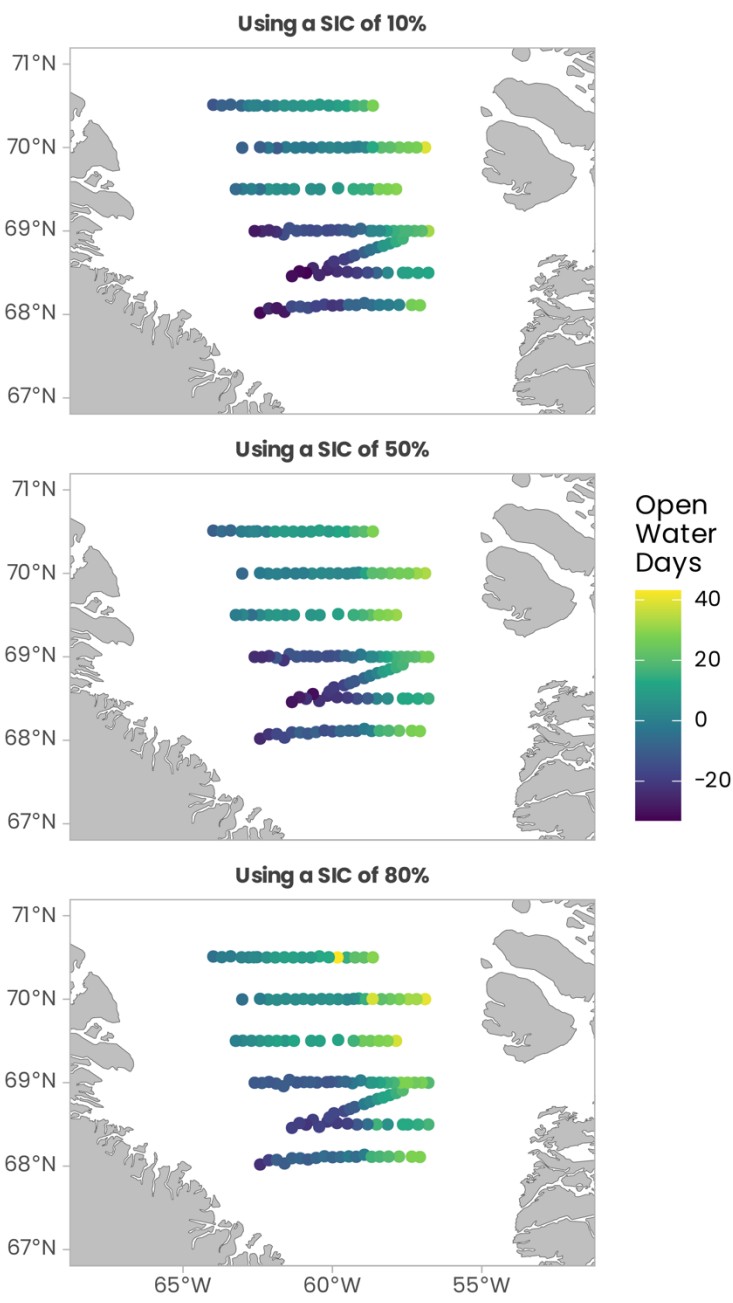

**Figure 7: Open water days (OWD) before sampling (values in days). The yellow colour corresponds to positive values meaning the water was already free of ice on the day of sampling. Bluer darker values correspond to stations that were still covered in ice at sampling date (negative values). Values of OWD can be computed using different SIC (sea-ice coverage) values: top panel SIC = 10%, middle panel SIC = 50% and bottom panel SIC = 80%.**

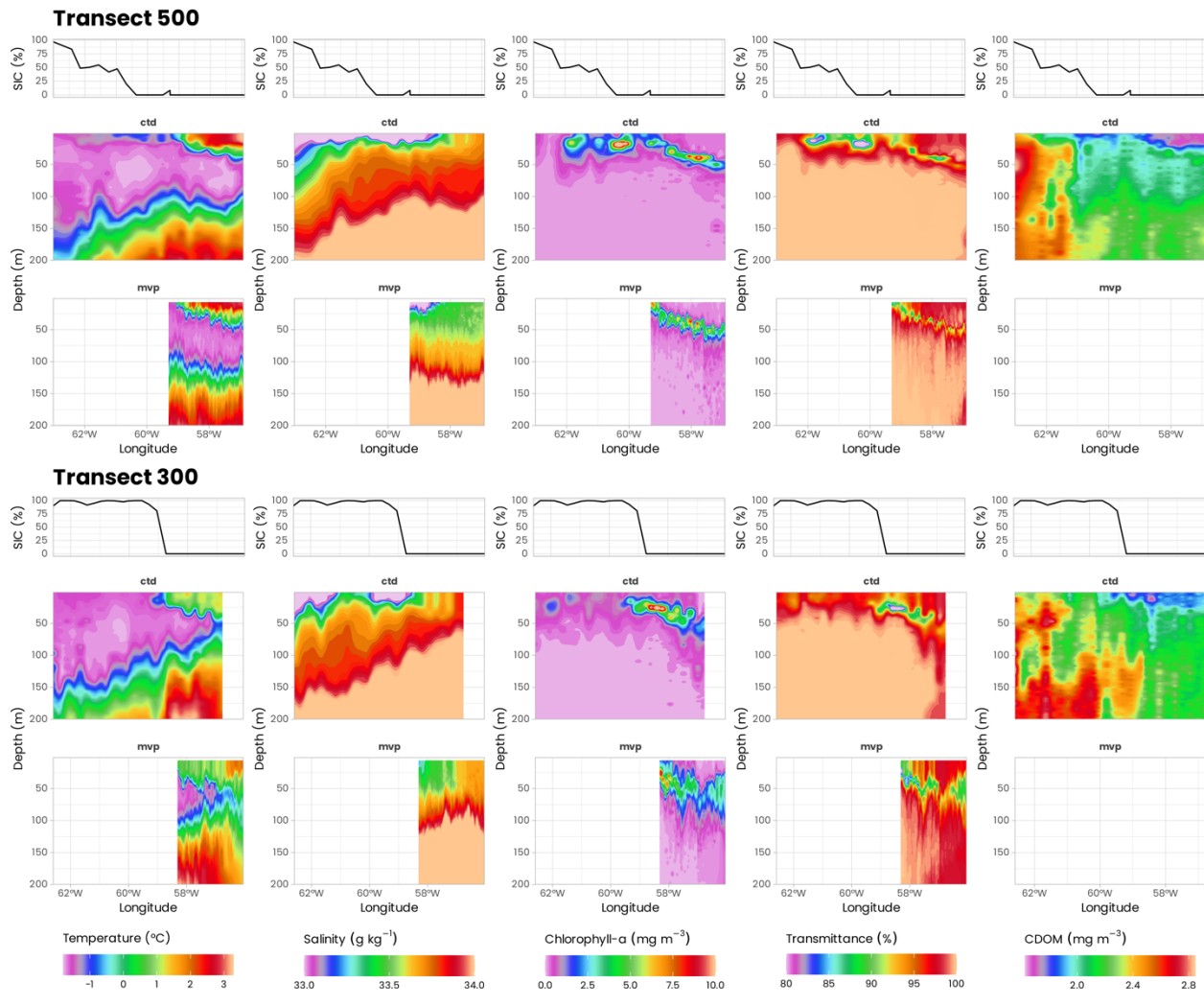

**Figure 8: Physical properties of seawater along transects 500 (70º North) and 300 (69º North). Middle and bottom panels show data recorded by sensors deployed on the rosette and on the MVP, respectively. From left to right: temperature (°C), salinity (g/Kg), chlorophyll-*a* concentration (mg m⁻³), transmittance (%) and CDOM concentration (mg m⁻³) as a function of depth (m, y axis) and longitude (degrees West, x axis). Topmost panels show ice coverage (%) at corresponding longitudes. Note that the MVP did not carry a CDOM sensor.**

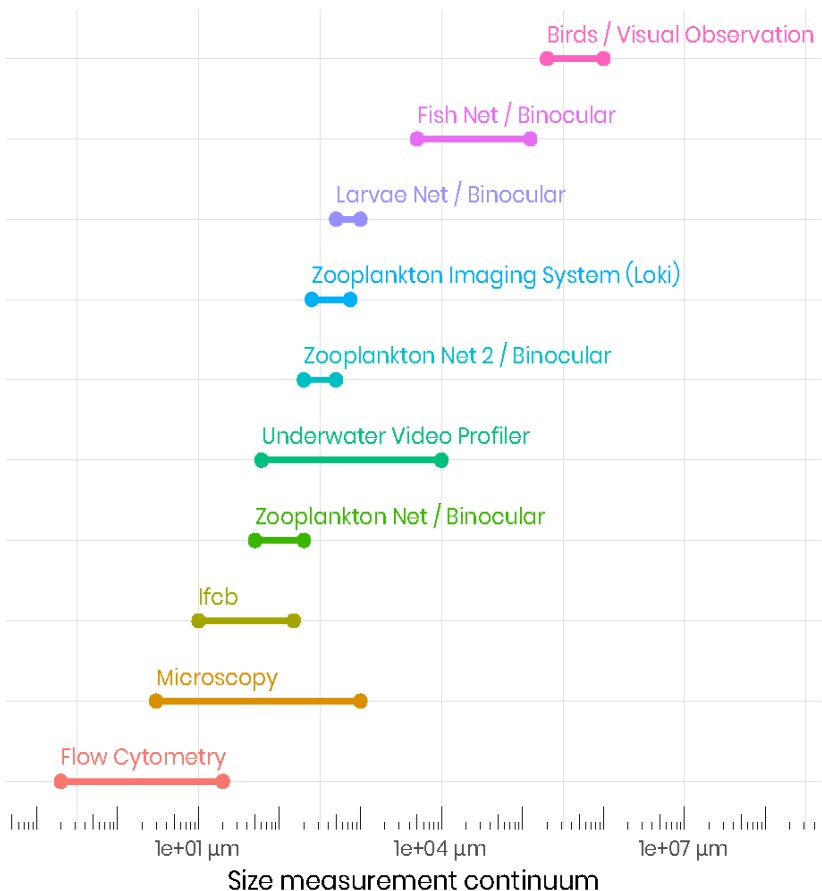

**Figure 9: Schematic of the biological sampling size continuum across the various methods and tools used during the Green Edge cruise.**

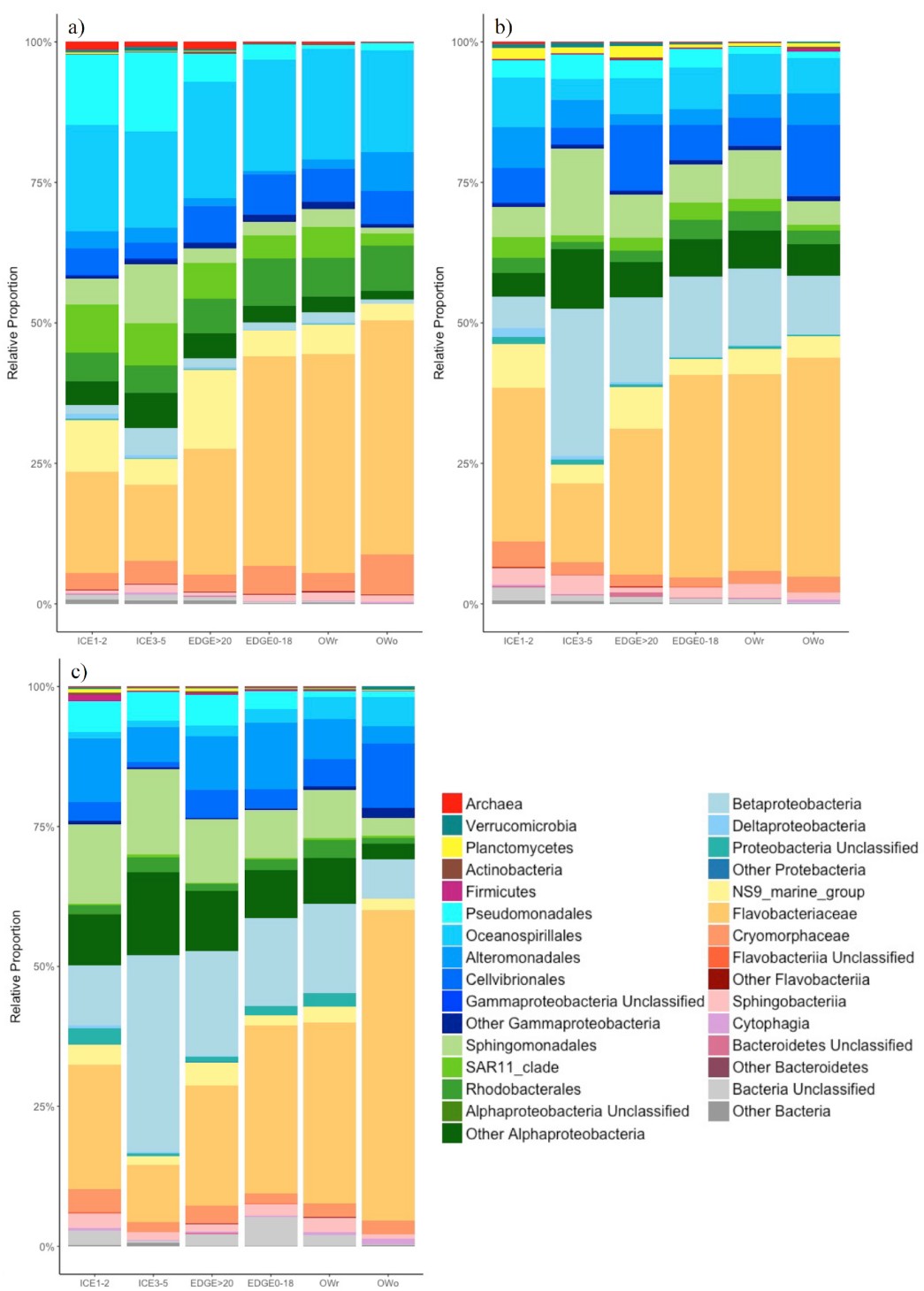

**Figure 10: Relative proportions of the different microbial taxa within the different groups of samples, a) for the 0.2-3 μm size fraction (free-living bacteria), b) for the 3-20 μm size fraction (particle-attached bacteria) and c) for the size fraction> 20 μm (particle-attached bacteria). ICE1-2: Ice stations in the transects 100 and 200; ICE3-5: Ice stations in transects 300 and 500; EDGE0-18: Samples between 0 and 18 m depth from the edge stations (stations 107, 204, and 312); EDGE> 20: Samples greater than 20 m deep from the edge stations (stations 107, 204, and 312); OWr: Open water stations where the ice had receded between 2 and 8 days previously; OWo: Open water stations where the ice has receded more than 15 days previously.**

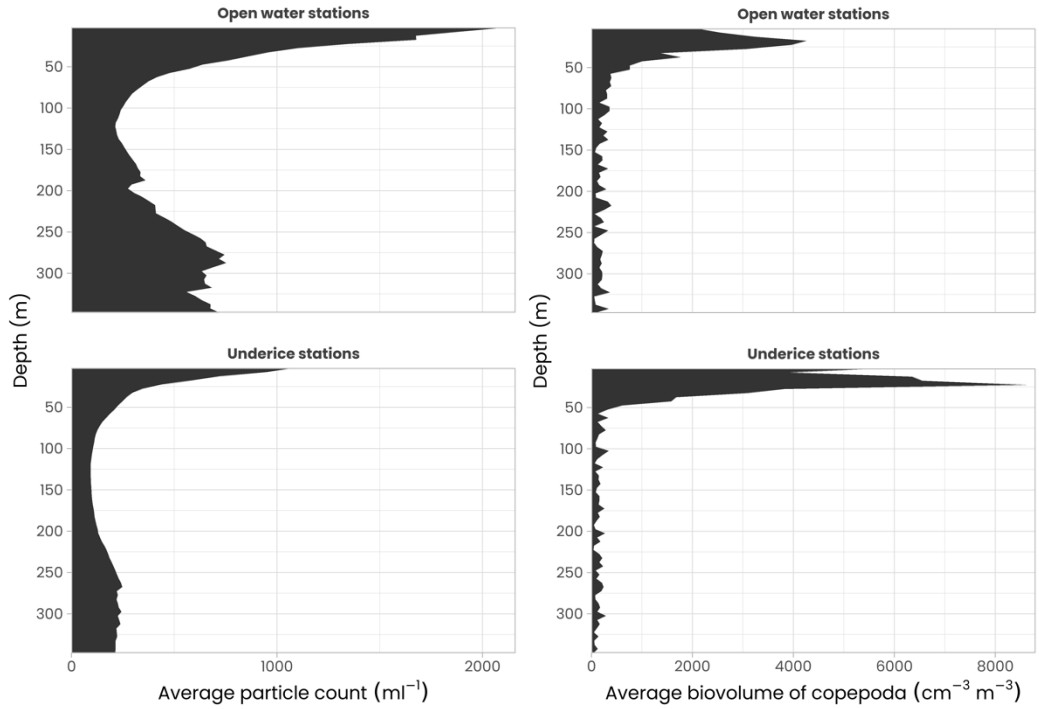

**Figure 11: Underwater Vision Profiler data. Average vertical profiles of particle concentration (per ml; left panels) and copepod volumetric fraction (cm³ m⁻³; right panels) at open-water (top panels) and ice-covered (bottom panels) stations over the top 350 m of the water column.**

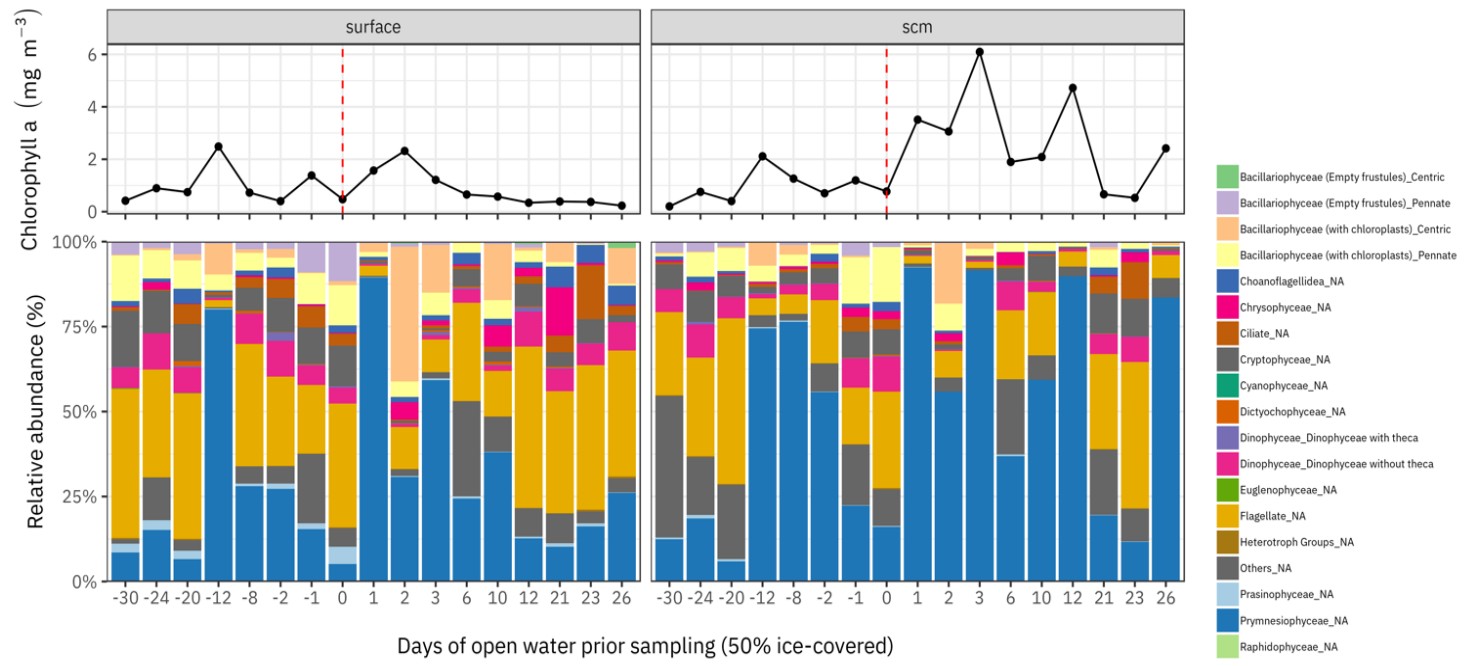

**Figure 12: Relative abundance (%) of main phytoplankton groups (colored bars) at the surface (left) and at the subsurface chlorophyll maximum (SCM; right) for all stations analyzed. Stations are sorted according to their OWD value (Fig. 7). Top panel represents the chlorophyll *a* concentration (mg m⁻³) at the relevant depths for each station.**

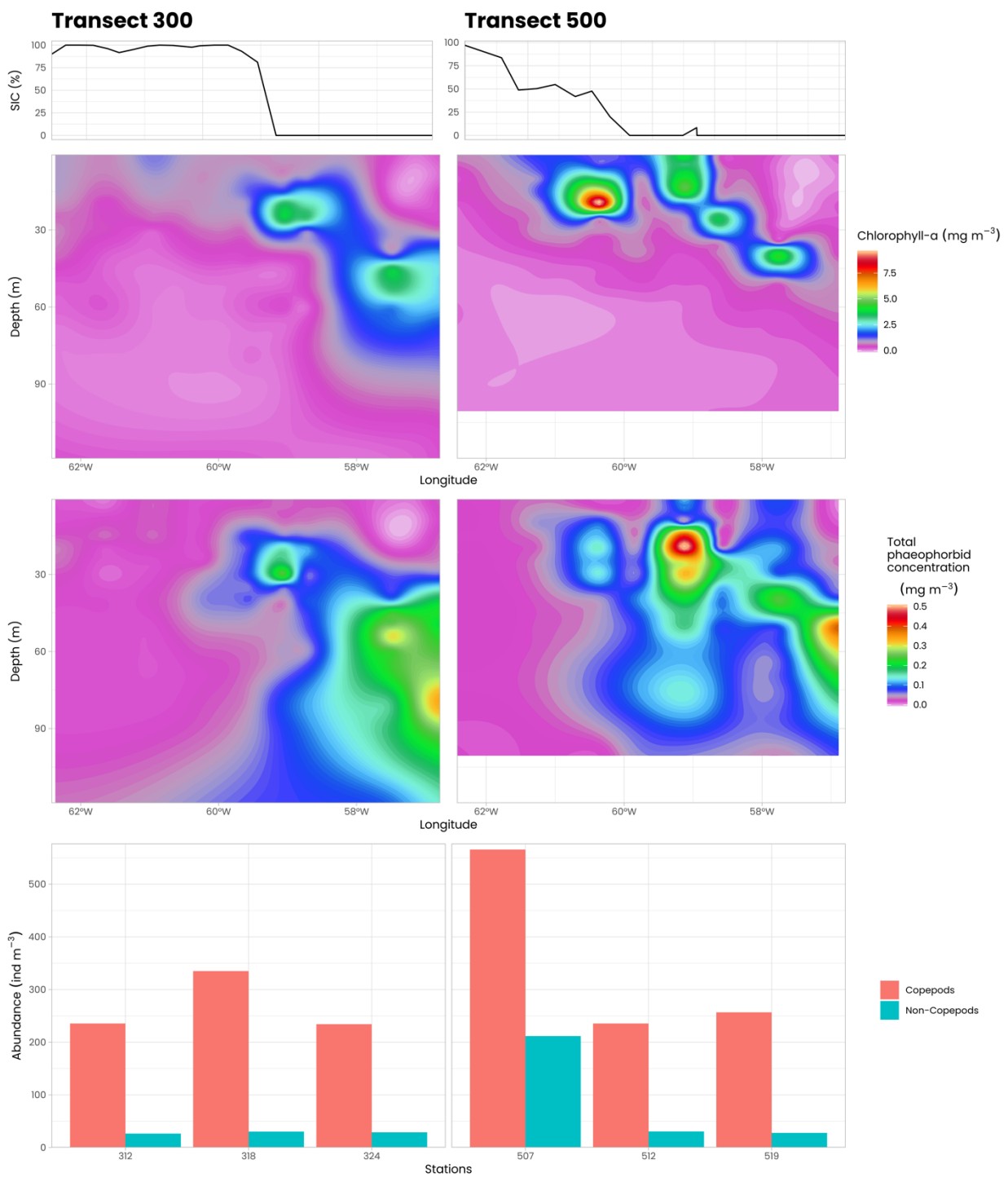

**Figure 13: Water column chlorophyll *a* concentration (mg m⁻³, top panels), and total phaeophorbide *a* concentration (mg m⁻³, bottom panels) along transects 300 (69º North, left panels) and 500 (70º North, right panels), as measured by HPLC on discrete samples obtained from the Niskin bottles at every FULL and BASIC station. Bottom panels, Abundance of zooplankton (ind m⁻³), copepods in red and non-copepods in blue, at each of the three FULL stations sampled on each transect (lower station numbers towards the east). The top graphs indicates the SIC at each station at the time of sampling.**

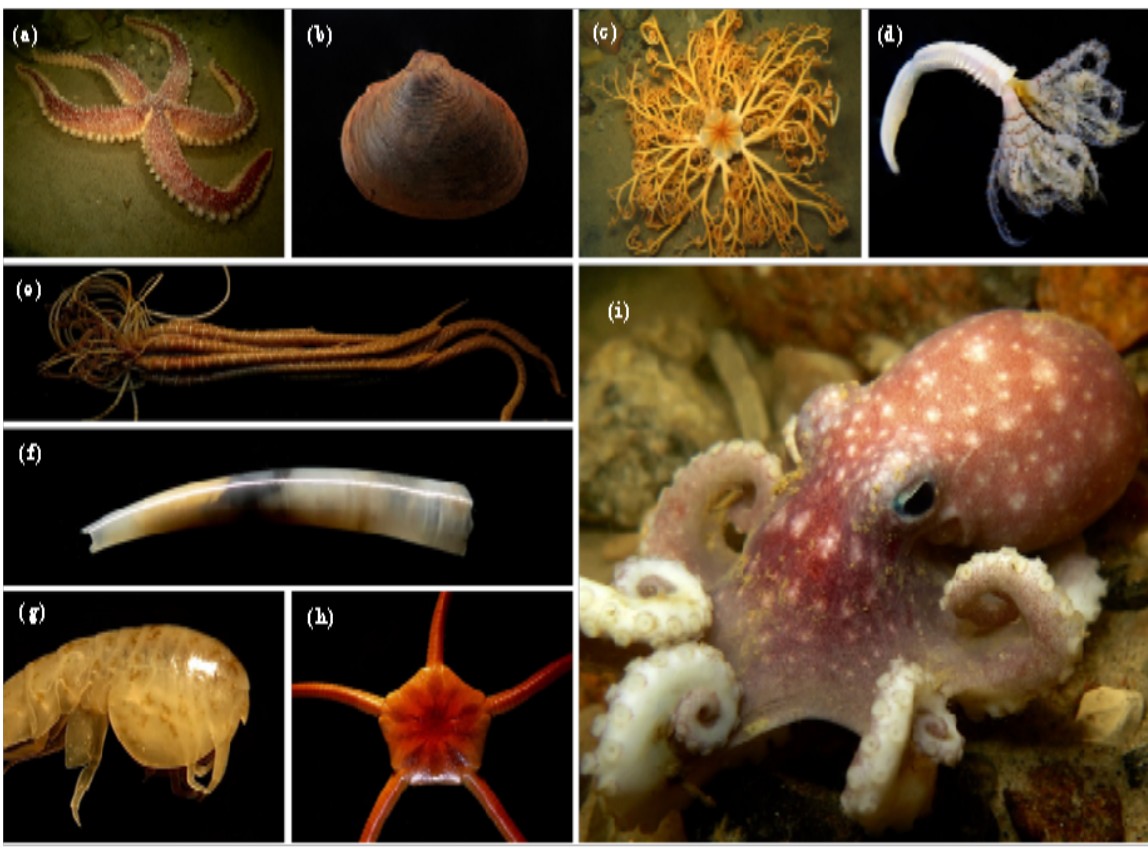

**Figure 14: Examples of benthic organisms. (a)** *Urasterias lincki*, **(b)** *Astarte borealis*, **(c)** *Gorgonocephalus eucnemis*, **(e)** *Brachiomma* **sp., (f)** *Heliometra glacialis*, **(g)** *Siphonodentalium lobatum*, **(h)** *Stegocephalus inflatus*, **(i)** *Ophiopleura borealis*, *Bathypolypus* **sp. (pictures by benthic ecology lab Université Laval, Gonzalo Bravo).**

**Table 1: List of operations carried out during the Green Edge cruise at each different station type. [(+) opportunistic sampling only]. CTD conductivity temperature depth, AOP apparent optical properties, C-OPS compact optical profiling system, IOP inherent optical properties, BB9 backscattering meter, CDOM colored dissolved organic matter, SCAMP self-contained autonomous microprofiler, LOKI lightframe onsight key species investigation.**

| OPERATION TYPE | FULL | BASIC | NUT | CTD |
|---|---|---|---|---|
| *CTD cast (see Table 2 for sensors list)* | + | + | + | + |
| *Water sampling (niskin/rosette)* | + | + | + | |
| *AOP profile (C-OPS)* | + | + | | |
| *IOP profile (CTD, BB9, a-Sphere, CDOM and EcoTriplet fluorometers)* | + | | | |
| *SCAMP profile* | + | | | |
| *LOKI and vertical integrated Plankton nets* | + | | | |
| *Mesozoic plankton trawl net* | + | | | |
| *Hydro-bios MultiNet Maxi* | + | | | |
| *Beam trawl* | + | (+) | | |
| *Agassiz trawl* | + | | | |
| *Box core* | + | (+) | | |
| *Ice sampling (when ice covered)* | + | | | |
| *Sediment traps* | *Deployed only once for long term sampling* | | | |
| TOTAL | 26 | 18 | 49 | 51 |

**Table 2: List of the sensors attached to the rosette carousel. CDOM colored dissolved organic matter, PAR photosynthetic available radiation, ADCP Acoustic doppler current profiler, UVP underwater vision profiler.**

| PARAMETER | SENSOR | UNIT | SERIAL # | CALIBRATION | # CASTS |
|---|---|---|---|---|---|
| *Temperature* | Sea-Bird Electronics SBE 3plus | ITS-90 deg C | 03P4318 | 12/2015 | 203 |
| *Conductivity* | Sea-Bird Electronics SBE 4 | mS/cm | 42696 | 12/2015 | 203 |
| *Pressure* | Paroscientific Digiquartz® | db | 679 | 01/2016 | 203 |
| *Oxygen Concentration* | Sea-Bird Electronics SBE 43 | mL / L | 430427 | 12/2015 | 203 |
| *CDOM fluorescence* | WetLabs ECO CDOM | mg / m$^3$ | 2344 | 01/2016 | 203 |
| *Chlorophyll fluorescence* | Seapoint SCF | µg / L | 3120 | 01/2016 | 203 |
| *Light attenuation* | WetLabs C-Star | % | CST671DR | 01/2016 | 87 |
| *Distance to bottom* | Benthos PSA-916 | m | 1065 | 02/2016 | 203 |
| *PAR / Irradiance* | Biospherical Instr. QCP-2300 | µEm$^{-2}$s$^{-1}$ | 4664 | 01/2014 | 203 |
| *Nitrate concentration* | Satlantic MBARI-ISUS | µM | 132 | 05/2016 | 104 |
| *Current speed and direction* | RDI-WHM300 L-ADCP | NA | | 03/2016 | 203 |
| *Particle concentration* | Hydroptic UVP5 | NA | | NA | 203 |

**Table 3. List of all the variables sampled during the Green Edge cruise.**

| VARIABLE | METHOD | SAMPLING METHOD | ACCESS TO DATASET | NAME OF FILE | PI |
|---|---|---|---|---|---|
| $^{15}N$-Nitrate assimilation | $^{15}N$ spiking - incubation - mass-spectrometry | Rosette Deck incubations | https://www.seanoe.org/data/00752/86417/ | Primary Production incubations experiments | Tremblay J.É. |
| $^{15}N$-Nitrate primary production ($^{13}C$) | $^{15}N$ spiking - incubation - mass-spectrometry | Rosette Deck incubations | https://www.seanoe.org/data/00752/86417/ | Primary Production incubations experiments | Tremblay J.É. |
| $^{15}N$-Urea assimilation | $^{15}N$ spiking - incubation - mass-spectrometry | Rosette Deck incubations | https://www.seanoe.org/data/00752/86417/ | Primary Production incubations experiments | Raimbault P. / Garcia N. |
| Above water Reflectance ($Rrs(0^+)$) | C-OPS Biospherical Instr. | Profile mode | https://www.seanoe.org/data/00752/86417/ | C-OPS data | Belanger S. |
| Above water Reflectance ($Rrs(0^+)$) | Radiometer (Satlantic HyperSAS) | Above-water sensor | https://www.seanoe.org/data/00752/86417/ | Remote Sensing Reflectance (Rrs) measured by the HyperSas | Belanger S. |
| Absorption (non-algal particles) | Spectrophotometer (filters) | Rosette water sample | https://www.seanoe.org/data/00752/86417/ | Particulate absorption | Bricaud A. / Sciandra A. / Matsuoka A. |
| Absorption (particulate matter) | Spectrophotometer (filters) | Rosette water sample | https://www.seanoe.org/data/00752/86417/ | Particulate absorption | Bricaud A. / Sciandra A. / Matsuoka A. |
| Absorption coefficient (total) | HOBI-Labs a-sphere | In-water profiler | http://www.obs-vlfr.fr/proof/ftpv/greenedge/db/DATA/AMUNDSEN/IOP/ | GreenEdge_hydroscat_Asphere | Belanger S. |
| Acoustic determination of fish presence | Echosounder | Continuous on way | https://www.polardata.ca/pdcsearch/?doi_id=12841 | Multiple files available. Data search required | Fortier L. |
| Air Relative Humidity | Humidity Sensor | Meteorological Tower | https://www.seanoe.org/data/00752/86417/ | Meteorological Tower data | Else B. / Burgers T. |
| Air Temperature | Temperature probe | Meteorological Tower | https://www.seanoe.org/data/00752/86417/ | Meteorological Tower data | Else B. / Burgers T. |
| Alkalinity total (TA) | Potentiometry | Rosette water sample | https://www.seanoe.org/data/00752/86417/ | Bottle salinity, DIC concentration and Delta 18O | Else B. / Miller L. |
| Ammonium (assimilation) | Isotopic dilution $^{15}N$ | Rosette Deck incubations | https://www.seanoe.org/data/00752/86417/ | Primary Production incubations experiments | Raimbault P. / Garcia N. |
| Ammonium (regeneration) | Isotopic dilution $^{15}N$ | Rosette Deck incubations | https://www.seanoe.org/data/00752/86417/ | Primary Production incubations experiments | Raimbault P. / Garcia N. |

| VARIABLE | METHOD | SAMPLING METHOD | ACCESS TO DATASET | NAME OF FILE | PI |
|---|---|---|---|---|---|
| *Ammonium concentration* | Fluorescence | Rosette water sample | https://www.seanoe.org/data/00752/86417/ | Water column nutrient concentrations | Tremblay J.É. |
| *Backscattering coefficient* | BGC-Argo profiler | In-water profiler | http://www.obs-vlfr.fr/proof/php/GREENEDGE/greenedge_autonomous.php | multiple | Babin M. |
| *Backscattering coefficient (6 wavelength)* | Hydroscat-6 | In-water profiler | http://www.obs-vlfr.fr/proof/ftpv/greenedge/db/DATA/AMUNDSEN/IOP/ | GreenEdge_hydroscat_Asphere | Belanger S. |
| *Bacterial abundance* | Flow Cytometry | Rosette water sample | https://www.seanoe.org/data/00752/86417/ | photosynthetic and non-photosynthetic eukaryotes and prokaryotes concentration (flow cytometry) | Vaulot D. |
| *Bacteria infected by virus* | Electron Microscopy | Rosette water sample | http://www.obs-vlfr.fr/proof/ftpv/greenedge/db/DATA/AMUNDSEN/EM/ | direct | Joux F. |
| *Bacterial diversity* | 16S rRNA Illumina sequencing | Rosette water sample | https://www.seanoe.org/data/00752/86417/ | Bacterial production and respiration | Joux F. |
| *Bacterial production* | Leucine-$^3$H incorporation | Rosette water sample | https://www.seanoe.org/data/00752/86417/ | bacterial production and respiration | Joux F. |
| *Bacterial respiration (whole community)* | $O_2$ consumption - Winkler - Incubations | Rosette water sample | https://www.seanoe.org/data/00752/86417/ | bacterial production and respiration | Joux F. |
| *Bacterial salinity- and light-induced biomarkers* | GC/MS | Rosette water sample | http://www.obs-vlfr.fr/proof/ftpv/greenedge/db/DATA/AMUNDSEN/Bact_Viab/ | direct | Rontani J.-F. / Amiraux R. / Burot C. |
| *Bacterial viability* | Incubation | Rosette water sample | http://www.obs-vlfr.fr/proof/ftpv/greenedge/db/DATA/AMUNDSEN/Bact_Viab/ | direct | Rontani J.-F. / Amiraux R. / Burot C. |
| *Bathymetry* | Kongsberg EM302 multi-beam echosounder | Continuous horizontal | https://geoapp.bibl.ulaval.ca/Home/Index | search | Lajeunesse P. / Joyal G. / Brouard E. |
| *Benthic ammonium flux* | Incubations - Colorimetry | Box corer | https://www.seanoe.org/data/00752/86417/ | Sediment incubations data | Morata N. |
| *Benthic carbon content* | CN analyser | Box corer | https://www.seanoe.org/data/00752/86417/ | Sediment incubations data | Morata N. |

| VARIABLE | METHOD | SAMPLING METHOD | ACCESS TO DATASET | NAME OF FILE | PI |
|---|---|---|---|---|---|
| *Benthic Macrofauna abundance* | Microscopy | Box corer | https://www.seanoe.org/data/00752/86417/ | Abundance and weight of benthic organisms | Archambault P. |
| *Benthic Macrofauna biomass* | Wet weight | Box corer | https://www.seanoe.org/data/00752/86417/ | Abundance and weight of benthic organisms | Archambault P. |
| *Benthic Macrofauna diversity* | Microscopy | Box corer | https://www.seanoe.org/data/00752/86417/ | Abundance and weight of benthic organisms | Archambault P. |
| *Benthic nitrate flux* | Incubations - Colorimetry-Autoanalyzer | Box corer | https://www.seanoe.org/data/00752/86417/ | Sediment incubations data | Morata N. |
| *Benthic nitrite flux* | Incubations - Colorimetry-Autoanalyzer | Box corer | https://www.seanoe.org/data/00752/86417/ | Sediment incubations data | Morata N. |
| *Benthic nitrogen content* | CN analyser | Box corer | https://www.seanoe.org/data/00752/86417/ | Sediment incubations data | Morata N. |
| *Benthic phosphate flux* | Incubations - Colorimetry-Autoanalyzer | Box corer | https://www.seanoe.org/data/00752/86417/ | Sediment incubations data | Morata N. |
| *Benthic silicic acid flux* | Incubations - Colorimetry - Autoanalyzer | Box corer | https://www.seanoe.org/data/00752/86417/ | Sediment incubations data | Morata N. |
| *Chlorophyll a and Phaeopigments concentration* | Fluorimetry | Sea ice core | https://www.seanoe.org/data/00752/86417/ | Chlorophyll and Phaeopigment concentration (fluorescence technique) | Bruyant F. / Babin M. |
| *Chlorophyll a and Phaeopigments concentration* | Fluorimetry | Rosette water sample | https://www.seanoe.org/data/00752/86417/ | Chlorophyll and Phaeopigment concentration (fluorescence technique) | Bruyant F. / Babin M. |
| *Chlorophyll a and Phaeopigments concentration (benthic)* | Fluorometric analysis | Box corer | http://www.obs-vlfr.fr/proof/ftpv/greenedge/db/DATA/AMUNDSEN/SEDIMENT_BC/ | GE_Amundsen_Sediment_BoxCore.csv | Archambault P. |
| *Chlorophyll a and Phaeopigments concentration (benthic)* | Fluorometric analysis | Box corer | https://www.seanoe.org/data/00752/86417/ | Sediment incubations data | Morata N. |
| *Chlorophyll a fluorescence* | Seapoint fluorometer | In-water profiler | https://www.seanoe.org/data/00752/86417/ | CTD data 2.0 m resolution with fluorescence | Guillot P. / Gombault C. |
| *Chlorophyll a fluorescence* | Fluorescence (Wetlabs) | In-water profiler | http://www.obs-vlfr.fr/proof/ftpv/greenedge/db/DATA/AMUNDSEN/IOP/ | GreenEdge_AMundsen_CTD_IOPs | Belanger S. |
| *Chlorophyll a fluorescence* | Fluorometer | Continuous horizontal | https://www.seanoe.org/data/00752/86417/ | CTD data 2.0m resolution with fluorescence | Guillot P. / Gombault C. |

| VARIABLE | METHOD | SAMPLING METHOD | ACCESS TO DATASET | NAME OF FILE | PI |
|---|---|---|---|---|---|
| *Chlorophyll a fluorescence* | BGC-Argo profiler | In-water profiler | http://www.obs-vlfr.fr/proof/php/GREENEDGE/greenedge_autonomous.php | multiple | Babin M. |
| *Chlorophyll a fluorescence* | Wetlabs FLRTD | Moving Vessel Profiler | https://www.seanoe.org/data/00752/86417/ | Moving vessel profiler data | Morisset S. / Gombault C. |
| *Chlorophyll-a concentration* | Fluorometer | Drifting sediment trap | https://www.seanoe.org/data/00752/86417/ | drifting traps data (25m depth) | Lalande C. |
| *Chromophoric Dissolved Organic Matter absorption* | Ultrapath | Rosette water sample | https://www.seanoe.org/data/00752/86417/ | Colored Disolved Organic Matter (CDOM) absorption data | Matsuoka A. |
| *Chromophoric Dissolved Organic Matter fluorescence* | Fluorescence (Wetlabs) | In-water profiler | http://www.obs-vlfr.fr/proof/ftpv/greenedge/db/DATA/AMUNDSEN/IOP/ | GreenEdge_AMundsen_CTD_IOPs | Belanger S. |
| *Chromophoric Dissolved Organic Matter fluorescence* | Fluorescence (Wetlabs, rosette) | In-water profiler | https://www.seanoe.org/data/00752/86417/ | CTD data 2.0m resolution with fluorescence | Guillot P. / Gombault C. |
| *Chromophoric Dissolved Organic Matter fluorescence* | BGC-Argo profiler | In-water profiler | http://www.obs-vlfr.fr/proof/php/GREENEDGE/greenedge_autonomous.php | multiple | Babin M. |
| *$CO_2$ partial pressure ($pCO_2$)* | Licor | Continuous horizontal | https://www.seanoe.org/data/00752/86417/ | Continuous $pCO_2$ and salinity | Else B. / Burgers T. |
| *Conductivity, Temperature, and depth (CTD)* | Seabird (IOP optical frame) | In-water profiler | http://www.obs-vlfr.fr/proof/ftpv/greenedge/db/DATA/AMUNDSEN/IOP/ | GreenEdge_AMundsen_CTD_IOPs | Belanger S. |
| *Conductivity, Temperature, and depth (CTD)* | Seabird (rosette) | In-water profiler | https://www.seanoe.org/data/00752/86417/ | CTD data 2.0m resolution with fluorescence | Guillot P. / Gombault C. |
| *Conductivity, Temperature, and depth (CTD)* | Seabird | Continuous horizontal | https://www.seanoe.org/data/00752/86417/ | CTD data 2.0m resolution with fluorescence | Guillot P. / Gombault C. |
| *Conductivity, Temperature, and depth (CTD)* | BGC-Argo profiler | In-water profiler | http://www.obs-vlfr.fr/proof/php/GREENEDGE/greenedge_autonomous.php | multiple | Babin M. |
| *Conductivity, Temperature, and depth (CTD)* | AML micro CTD | Moving Vessel Profiler | https://www.seanoe.org/data/00752/86417/ | Moving vessel profiler data | Morisset S. / Gombault C. |

| VARIABLE | METHOD | SAMPLING METHOD | ACCESS TO DATASET | NAME OF FILE | PI |
|---|---|---|---|---|---|
| *Cryptophytes (abundance)* | Flow cytometry | Rosette water sample | https://www.seanoe.org/data/00752/86417/ | photosynthetic and non-photosynthetic eukaryotes and prokaryotes concentration (flow cytometry) | Vaulot D. |
| *Current speed and direction* | ADCP (150kHz) | Continuous horizontal | https://www.seanoe.org/data/00752/86417/ | Acoustic Doppler Current Profiler (ADCP) | Guillot P. / Gombault C. |
| *Current speed and direction* | ADCP (LADCP) | In-water profiler | https://www.seanoe.org/data/00752/86417/ | Acoustic Doppler Current Profiler (ADCP) | Guillot P. / Gombault C. |
| *delta $^{18}O$ - water* | Mass Spectrometry | Rosette water sample | https://www.seanoe.org/data/00752/86417/ | Bottle salinity, DIC concentration and Delta $^{18}O$ | Else B. / Mucci A. |
| *Demersal fish diversity* | Beam trawl | Fish trawl | https://www.seanoe.org/data/00752/86417/ | Demersal fish abundance and sizes (Beam trawl sampling) | Fortier L. |
| *Diacids (Aerosol)* | HPAEC-PAD | Atmosphere | http://www.obs-vlfr.fr/proof/ftpv/greenedge/db/DATA/AMUNDSEN/AEROSOLS/ | direct | Panagiotopoulos C. / Sempere R. |
| *Diatom (abundance)* | Microscopy | Rosette water sample | https://www.seanoe.org/data/00752/86417/ | Taxonomy data Diatoms abundance inverted microscope | Lafond A. |
| *Diatom frustules (abundance)* | Ludox/colloidal silica extraction | Box corer | https://www.seanoe.org/data/00752/86417/ | Sediment incubations data | Morata N. |
| *Diatoms (bacilliarophyta) abundance* | Inverted microscopy | Rosette water sample | https://www.seanoe.org/data/00752/86417/ | Taxonomy data Diatoms abundance inverted microscope | Leblanc K. / Quéguiner B./ Cornet V. |
| *Diatoms (bacilliarophyta) taxonomy* | Inverted microscopy | Rosette water sample | https://www.seanoe.org/data/00752/86417/ | Taxonomy data Diatoms abundance inverted microscope | Leblanc K. / Quéguiner B./ Cornet V. |
| *Diffuse attenuation coefficient ($K_d$)* | C-OPS | In-water profiler | https://www.seanoe.org/data/00752/86417/ | C-OPS data | Belanger S. |
| *Dimethyl sulfide (DMS)* | Gas Chromatography-Mass Spectrometry | Rosette water sample | https://www.seanoe.org/data/00752/86417/ | DMS and DMSP concentration | Masse G. / Galí M. |
| *Dimethyl sulfide (sea-air flux)* | Gas Chromatography-Mass Spectrometry | Rosette water sample | https://www.seanoe.org/data/00752/86417/ | DMS and DMSP concentration | Masse G. / Galí M. |

| VARIABLE | METHOD | SAMPLING METHOD | ACCESS TO DATASET | NAME OF FILE | PI |
|---|---|---|---|---|---|
| *Dimethylsulfopropionate (DMSP)* | Gas Chromatography-Mass Spectrometry | Rosette water sample | https://www.seanoe.org/data/00752/86417/ | DMS and DMSP concentration | Masse G. / Galí M. / Lizotte M. / Hussherr R. |
| *Dissolved inorganic Carbon (DIC)* | Coulometry | Rosette water sample | https://www.seanoe.org/data/00752/86417/ | Bottle salinity, DIC concentration and Delta $^{18}$O | Else B. / Miller L. |
| *Dissolved Organic Carbon (DOC)* | Wet oxidation | Rosette Deck incubations | https://www.seanoe.org/data/00752/86417/ | Dissolved Inorganic and Organic matter concentrations | Raimbault P. / Garcia N. |
| *Dissolved Organic Nitrogen* | Wet oxidation | Rosette Deck incubations | https://www.seanoe.org/data/00752/86417/ | Dissolved Inorganic and Organic matter concentrations | Raimbault P. / Garcia N. |
| *Dissolved Organic Nitrogen (release)* | Isotopic procedure | Rosette Deck incubations | https://www.seanoe.org/data/00752/86417/ | Primary Production incubations experiments | Raimbault P. / Garcia N. |
| *Dissolved Oxygen concentration* | Seabird 43 | In-water profiler | https://www.seanoe.org/data/00752/86417/ | CTD data 2.0m resolution with fluorescence | Guillot P. / Gombault C. |
| *Dissolved Oxygen concentration* | BGC-Argo profiler | In-water profiler | http://www.obs-vlfr.fr/proof/php/GREENEDGE/greenedge_autonomous.php | multiple | Babin M. |
| *Downward longwave radiation* | Pyrgeometer | Atmosphere | https://www.seanoe.org/data/00752/86417/ | Downwelling radiation (pyrgeometer) | Else B. / Burgers T. |
| *Downward shortwave radiation* | Pyranometer | Atmosphere | https://www.seanoe.org/data/00752/86417/ | Downwelling radiation (pyrgeometer) | Else B. / Burgers T. |
| *Downwelling Irradiance ($E_d(z)$)* | C-OPS | In-water profiler | https://www.seanoe.org/data/00752/86417/ | C-OPS data | Belanger S. |
| *Downwelling irradiance above the surface ($E_d(0^+)$)* | Radiometer (Satlantic HyperSAS) | Above-water sensor | https://www.seanoe.org/data/00752/86417/ | Remote Sensing Reflectance ($R_{rs}$) measured by the HyperSas | Belanger S. |
| *Downwelling Irradiance above the surface ($E_d(0^+)$)* | SBDART | Surface mode | http://www.obs-vlfr.fr/proof/ftpv/greenedge/db/DATA/SBDART/AM2016/ | AM2016_SBDART_AllCasts.zip | Babin M. / Galí M. |
| *Downwelling Irradiance above the surface ($E_d(0^+)$)* | C-OPS | In-water profiler | https://www.seanoe.org/data/00752/86417/ | C-OPS data | Belanger S. |

| VARIABLE | METHOD | SAMPLING METHOD | ACCESS TO DATASET | NAME OF FILE | PI |
|---|---|---|---|---|---|
| *Downwelling Radiance ($E_d(z)$)* | BGC-Argo profiler | In-water profiler | http://www.obs-vlfr.fr/proof/php/GREENEDGE/greenedge_autonomous.php | multiple | Babin M. |
| *Epibenthic fauna abundance* | Microscopy | Agassiz trawl | https://www.seanoe.org/data/00752/86417/ | Benthic organisms' identification and abundance | Archambault P. |
| *Epibenthic fauna biomass* | Wet weight | Agassiz trawl | https://www.seanoe.org/data/00752/86417/ | Benthic organisms' identification and abundance | Archambault P. |
| *Epibenthic fauna diversity* | Microscopy | Agassiz trawl | https://www.seanoe.org/data/00752/86417/ | Benthic organisms' identification and abundance | Archambault P. |
| *Eukaryotic diversity* | Metabarcoding | Rosette water sample | https://www.seanoe.org/data/00752/86417/ | photosynthetic and non-photosynthetic eukaryotes and prokaryotes concentration (flow cytometry) | Vaulot D. |
| *Fish abundance (Midwater)* | IKMT trawl | Fish trawl | https://www.seanoe.org/data/00752/86417/ | Pelagic fish abundance and sizes (IKMT sampling) | Fortier L. |
| *Lipid biomarkers concentrations* | GC/MS | Collected Organisms (bird) | https://www.seanoe.org/data/00752/86417/ | Lipid biomarkers in benthic and sediment fauna | Mosbech A. / Fort J. |
| *Microturbulence* | SCAMP profiler | In-water profiler | http://www.obs-vlfr.fr/proof/ftpv/greenedge//db/DATA/AMUNDSEN/SCAMP/ | | Vladoiu A. / Dumont D. |
| *Nanoeukaryotes (abundance)* | Flow cytometry | Rosette water sample | https://www.seanoe.org/data/00752/86417/ | photosynthetic and non-photosynthetic eukaryotes and prokaryotes concentration (flow cytometry) | Vaulot D. |
| *Nitrate ($NO_3^-$) assimilation* | Isotopic dilution 15N | Deck incubations | https://www.seanoe.org/data/00752/86417/ | Primary Production incubations experiments | Raimbault P. / Garcia N. |
| *Nitrate concentration ($NO_3^-$)* | Colorimetry/Autoanalyser | Rosette water sample | https://www.seanoe.org/data/00752/86417/ | Water column nutrient concentrations | Tremblay J. E. |
| *Nitrate concentration ($NO_3^-$)* | Colorimetry/Autoanalyser | Deck incubations | https://www.seanoe.org/data/00752/86417/ | Primary production incubations experiments | Garcia N. |
| *Nitrate concentration ($NO_3^-$)* | BGC-Argo profiler | In-water profiler | http://www.obs-vlfr.fr/proof/php/GREENEDGE/greenedge_autonomous.php | multiple | Babin M. |

| VARIABLE | METHOD | SAMPLING METHOD | ACCESS TO DATASET | NAME OF FILE | PI |
|---|---|---|---|---|---|
| *Nitrate concentration (NO₃)* | MBARI-ISUS Satlantic | In-water profiler | https://www.seanoe.org/data/00752/86417/ | CTD data 2.0m resolution with fluorescence | Guillot P. / Gombault C. |
| *Nitrification* | $^{15}$N labeling | Deck incubations | https://www.seanoe.org/data/00752/86417/ | Primary Production incubations experiments | Raimbault P. / Garcia N. |
| *Nitrite concentration (NO₂⁻)* | Colorimetry/Autoanalyser | Rosette water sample | https://www.seanoe.org/data/00752/86417/ | Water column nutrient concentrations | Tremblay J. E. |
| *Nitrite concentration (NO₂⁻)* | Colorimetry/Autoanalyser | Deck incubations | https://www.seanoe.org/data/00752/86417/ | Primary production incubations experiments | Garcia N. |
| *Open Water Days* | AMSR | Satellite | https://www.seanoe.org/data/00752/86417/ | Days of Open Water (DOW) | Massicotte P. / Ferland J. |
| *Orthosilicic acid (uptake rate)* | $^{32}$Si absorption | Rosette water sample | https://www.seanoe.org/data/00752/86417/ | Silicate uptake rate | Leynaert A./ Quéguiner B./ Gallinari M. |
| *Orthosilicic acid concentration (Si(OH)₄)* | Colorimetry/Autoanalyser | Rosette water sample | https://www.seanoe.org/data/00752/86417/ | Water column nutrient concentrations | Tremblay J.É. |
| *Orthosilicic acid concentration (Si(OH)₄)* | Colorimetry/Autoanalyser | Deck incubations | https://www.seanoe.org/data/00752/86417/ | Primary production incubations experiments | Garcia N. |
| *Orthosilicic acid concentration Si(OH)₄* | Technicon | Rosette water sample | https://www.seanoe.org/data/00752/86417/ | Silicate absorption kinetics experiments | Leynaert A. / Moriceau B./ Gallinari M. |
| *Orthosilicic acid Si(OH)₄ - uptake kinetics* | $^{32}$Si absorption - incubation | Rosette water sample | https://www.seanoe.org/data/00752/86417/ | Silicate absorption kinetics experiments | Leynaert A./ Quéguiner B./ Gallinari M. |
| *Particle Size Distribution* | UVP-5 | In-water profiler | https://www.seanoe.org/data/00752/86417/ | Underwater Vision Profiler (UVP) Particles | Picheral M. |
| *Particulate mass* | Dry Weight | Drifting sediment trap | https://www.seanoe.org/data/00752/86417/ | drifting traps data (25m depth) | Lalande C. |
| *Particulate Nitrogen content* | CHN | Drifting sediment trap | https://www.seanoe.org/data/00752/86417/ | drifting traps data (25m depth) | Lalande C. |
| *Particulate Organic Carbon (POC)* | CHN | Rosette water sample | https://www.seanoe.org/data/00752/86417/ | Dry weight, Particulate Carbon and Nitrogen (CHN) | Bruyant F. / Lariviere J. / Babin M. |
| *Particulate Organic Carbon (POC)* | High combustion | Deck incubations | https://www.seanoe.org/data/00752/86417/ | Dissolved Inorganic and Organic matter concentrations | Raimbault P. / Garcia N. |

| VARIABLE | METHOD | SAMPLING METHOD | ACCESS TO DATASET | NAME OF FILE | PI |
|---|---|---|---|---|---|
| *Particulate Organic Carbon (POC)* | CHN | Drifting sediment trap | https://www.seanoe.org/data/00752/86417/ | drifting traps data (25m depth) | Lalande C. |
| *Particulate Organic Nitrogen (PON)* | CHN | Rosette water sample | https://www.seanoe.org/data/00752/86417/ | Dry weight, Particulate Carbon and Nitrogen (CHN) | Bruyant F. / Lariviere J. / Babin M. |
| *Particulate Organic Nitrogen (PON)* | High combustion | Deck incubations | https://www.seanoe.org/data/00752/86417/ | Dissolved Inorganic and Organic matter concentrations | Raimbault P. / Garcia N. |
| *PDMPO uptake* | Spectrophotometry/Spectrofluorometry | Rosette water sample | https://www.seanoe.org/data/00752/86417/ | Silicification of Diatoms | Leblanc K. / Quéguiner B./ Cornet V. |
| *Phosphate concentration $(PO_4)^{3-})$* | Colorimetry/Autoanalyser | Rosette water sample | https://www.seanoe.org/data/00752/86417/ | Water column nutrient concentrations | Tremblay J.É. |
| *Phosphate concentration $(PO_4)^{3-})$* | Colorimetry/Autoanalyser | Deck incubations | https://www.seanoe.org/data/00752/86417/ | Primary production incubations experiments | Garcia N. |
| *Photo Eukaryotes (morphology)* | Scanning Electron Microscopy | Rosette water sample | https://www.seanoe.org/data/00752/86417/ | photosynthetic and non-photosynthetic eukaryotes and prokaryotes concentration (flow cytometry) | Vaulot D. |
| *Photo Eukaryotes Sorted (Morphology)* | Scanning Electron Microscopy | Rosette water sample | https://www.seanoe.org/data/00752/86417/ | photosynthetic and non-photosynthetic eukaryotes and prokaryotes concentration (flow cytometry) | Vaulot D. |
| *Photosynthetic Available Radiation (PAR)* | Biospherical Instrument QCP-2300 | In-water profiler | https://www.seanoe.org/data/00752/86417/ | CTD data 2.0m resolution with fluorescence | Guillot P. / Gombault C. |
| *Photosynthetic Available Radiation (PAR)* | C-OPS | In-water profiler | https://www.seanoe.org/data/00752/86417/ | C-OPS data | Belanger S. |
| *Photosynthetic Available Radiation (PAR)* | SBDART | Surface mode | http://www.obs-vlfr.fr/proof/ftpv/greenedge/db/DATA/SBDART/AM2016/ | AM2016_SBDART_AllCasts.zip | Babin M. / Galí M. |
| *Photosynthetic Available Radiation (PAR)* | Radiometer | Atmosphere | https://www.seanoe.org/data/00752/86417/ | Downwelling radiation (pyrgeometer) | Else B. / Burgers T. |

| VARIABLE | METHOD | SAMPLING METHOD | ACCESS TO DATASET | NAME OF FILE | PI |
|---|---|---|---|---|---|
| *Photosynthetic Available Radiation (PAR)* | BGC-Argo profiler | In-water profiler | http://www.obs-vlfr.fr/proof/php/GREENEDGE/greenedge_autonomous.php | multiple | Babin M. |
| *Photosynthetic parameters* | $^{14}$C P vs. E curve | Rosette water sample | https://www.seanoe.org/data/00752/86417/ | Photosynthetic parameters | Lewis K. |
| *Phytoplankton (taxonomy)* | Inverted Microscopy | Rosette water sample | | Phytoplankton taxonomy (microscopy) | Babin M. |
| *Phytoplankton (Micro-) taxonomy* | Microscopy | Drifting sediment trap | https://www.seanoe.org/data/00752/86417/ | drifting traps data (25m depth) | Lalande C. |
| *Phytoplankton (taxonomy)* | Imaging flowcytobot | Sea ice core | http://www.obs-vlfr.fr/proof/ftpv/greenedge/db/DATA/AMUNDSEN/IFCB/ | multiple | Bruyant F. / Grondin P.L. / Babin M. |
| *Phytoplankton (taxonomy)* | Imaging flowcytobot | Rosette water sample | http://www.obs-vlfr.fr/proof/ftpv/greenedge/db/DATA/AMUNDSEN/IFCB/ | multiple | Bruyant F. / Grondin P.L. / Babin M. |
| *Phytoplankton cultures* | Sorted by flow cytometry, serial dilution, single cell pipetting | Rosette water sample | https://www.seanoe.org/data/00752/86417/ | photosynthetic and non-photosynthetic eukaryotes and prokaryotes concentration (flow cytometry) | Vaulot D. |
| *Picoeukaryotes (abundance)* | Flow cytometry | Rosette water sample | https://www.seanoe.org/data/00752/86417/ | photosynthetic and non-photosynthetic eukaryotes and prokaryotes concentration (flow cytometry) | Vaulot D. |
| *Pigments concentration* | HPLC | Rosette water sample | https://www.seanoe.org/data/00752/86417/ | phytoplankton pigments concentration (HPLC) | Ras J. / Claustre H. / Dimier C. |
| *Pigments concentration* | HPLC | Sea ice core | https://www.seanoe.org/data/00752/86417/ | phytoplankton pigments concentration (HPLC) | Ras J. / Claustre H. / Dimier C. |
| *Plankton taxonomy* | UVP-5 | In-water profiler | https://www.seanoe.org/data/00752/86417/ | Underwater Vision Profiler (UVP) zooplankton | Picheral M. |
| *Primary production* | $^{13}$C labeling | Deck incubations | https://www.seanoe.org/data/00752/86417/ | Primary Production incubations experiments | Raimbault P. / Garcia N. |
| *Quantum Efficiency of PSII (Fv/Fm) (Phytoplankton)* | Benchtop PAM - PhytoPAM | Rosette water sample | http://www.obs-vlfr.fr/proof/ftpv/greenedge/db/DATA/AMUNDSEN/PAM/ | direct | Joy-Warren H. |

| VARIABLE | METHOD | SAMPLING METHOD | ACCESS TO DATASET | NAME OF FILE | PI |
|---|---|---|---|---|---|
| *Sea ice concentration* | Satellite | Surface mode | https://www.seanoe.org/data/00752/86417/ | Sea ice history | Massicotte P. |
| *Sediment grain size* | Laser | Box corer | http://www.obs-vlfr.fr/proof/ftpv/greenedge/db/DATA/AMUNDSEN/SEDIMENT_BC/ | GE_Amundsen_Sediment_BoxCore.csv | Archambault P. |
| *Sediment organic content* | Lost by ignition | Box corer | http://www.obs-vlfr.fr/proof/ftpv/greenedge/db/DATA/AMUNDSEN/SEDIMENT_BC/ | GE_Amundsen_Sediment_BoxCore.csv | Archambault P. |
| *Silica Biogenic (BSi)* | BSi extraction | Rosette water sample | https://www.seanoe.org/data/00752/86417/ | Biogenic and Lithogenic silica concentration | Leblanc K./ Quéguiner B./ Leynaert A./ Moriceau B./ Legras J./ Gallinari M. |
| *Silica Biogenic (BSi) dissolution rate* | dissolution in filtered sea water | Rosette water sample | https://www.seanoe.org/data/00752/86417/ | Biogenic and Lithogenic silica concentration | Moriceau B./ Gallinari M. |
| *Silica Biogenic concentration* | BSi extraction | Box corer | https://www.seanoe.org/data/00752/86417/ | Sediment incubations data | Morata N./ Gallinari M. |
| *Silica Lithogenic (LSi)* | LSi extraction | Rosette water sample | https://www.seanoe.org/data/00752/86417/ | Biogenic and Lithogenic silica concentration | Leblanc K./ Quéguiner B./ Leynaert A./ Moriceau B./ Legras J./ Gallinari M. |
| *Sub-bottom profiles* | Knudsen 320 sub-bottom echosounder | Continuous horizontal | https://geoapp.bibl.ulaval.ca/Home/Index | search | Lajeunesse P. / Joyal G. / Brouard E. |
| *Sugars (Aerosol)* | HPAEC-PAD | Atmosphere | http://www.obs-vlfr.fr/proof/ftpv/greenedge/db/DATA/AMUNDSEN/AEROSOLS/ | direct | Panagiotopoulos C. / Sempere R. |
| *Suspended Particulate Material (SPM)* | Particle dry weight ( gravimetry) | Rosette water sample | https://www.seanoe.org/data/00752/86417/ | Dry weight, Particulate Carbon and Nitrogen (CHN) | Bruyant F. / Lariviere J. / Babin M. |

| VARIABLE | METHOD | SAMPLING METHOD | ACCESS TO DATASET | NAME OF FILE | PI |
|---|---|---|---|---|---|
| *Synechococcus (abundance)* | Flow cytometry | Rosette water sample | https://www.seanoe.org/data/00752/86417/ | photosynthetic and non-photosynthetic eukaryotes and prokaryotes concentration (flow cytometry) | Vaulot D. |
| *Total Organic Carbon (TOC)* | Wet oxidation | Rosette Deck incubations | https://www.seanoe.org/data/00752/86417/ | Dissolved Inorganic and Organic matter concentrations | Raimbault P. / Garcia N. |
| *Total Organic Nitrogen (TON)* | Wet oxidation | Rosette Deck incubations | https://www.seanoe.org/data/00752/86417/ | Dissolved Inorganic and Organic matter concentrations | Raimbault P. / Garcia N. |
| *Total Organic Phosphorus (TOP)* | Wet oxidation | Rosette Deck incubations | https://www.seanoe.org/data/00752/86417/ | Dissolved Inorganic and Organic matter concentrations | Raimbault P. / Garcia N. |
| *Transmittance (of light in water)* | CST Wetlabs | Moving Vessel Profiler | https://www.seanoe.org/data/00752/86417/ | Moving vessel profiler data | Morisset S. / Gombault C. |
| *Transmittance (of light in water)* | C-Star | In-water profiler | https://www.seanoe.org/data/00752/86417/ | CTD data 2.0m resolution with fluorescence | Guillot P. / Gombault C. |
| *Upwelling Irradiance ($E_u(z)$)* | C-OPS | In-water profiler | https://www.seanoe.org/data/00752/86417/ | C-OPS data | Belanger S. |
| *Upwelling radiance ($L_u$)* | C-OPS | In-water profiler | https://www.seanoe.org/data/00752/86417/ | C-OPS data | Belanger S. |
| *Variable fluorescence and Rapid Light Curves parameters (Phytoplankton)* | Benchtop PAM | Rosette water sample | http://www.obs-vlfr.fr/proof/ftpv/greenedge/db/DATA/AMUNDSEN/PAM/ | direct | Joy-Warren H. |
| *Weather and navigation* | On Board | AVOS | https://www.seanoe.org/data/00752/86417/ | Automated Volunteer Observing Ship (AVOS) weather data | Morisset S. / Gombault C. |
| *Wind Direction* | Wind monitor | Meteorological Tower | https://www.seanoe.org/data/00752/86417/ | Meteorological Tower data | Else B. / Burgers T. |
| *Wind Speed* | Wind monitor | Meteorological Tower | https://www.seanoe.org/data/00752/86417/ | Meteorological Tower data | Else B. / Burgers T. |
| *Wind speed* | Cross Calibrated Multi Platform (CCMP) | Atmosphere | http://www.obs-vlfr.fr/proof/ftpv/greenedge//db/DATA/WIND/DATA/ | direct | Massicotte P. |

| VARIABLE | METHOD | SAMPLING METHOD | ACCESS TO DATASET | NAME OF FILE | PI |
|---|---|---|---|---|---|
| *zooplankton (abundance)* | Multi nets (Hydro-bios) | Plankton net | https://www.seanoe.org/data/00752/86417/ | Zooplankton abundance and diversity (vertical nets) | Fortier L. |
| *Zooplankton (Abundances)* | Vertical plankton net 200 µm | Plankton net | https://www.seanoe.org/data/00752/86417/ | Zooplankton abundance and diversity (vertical nets) | Fortier L. |
| *Zooplankton (Meso-) (abundance)* | Tucker net | Plankton net | https://www.seanoe.org/data/00752/86417/ | Ichthyoplancton vertical nets sampling | Fortier L. |
| *Zooplankton (Meso-) (taxonomy)* | Tucker net | Plankton net | https://www.seanoe.org/data/00752/86417/ | Ichthyoplancton vertical nets sampling | Fortier L. |
| *Zooplankton (taxonomy)* | Multi nets (Hydro-bios) | Plankton net | https://www.seanoe.org/data/00752/86417/ | Zooplankton abundance and diversity (vertical nets) | Fortier L. |
| *Zooplankton (Taxonomy)* | Vertical plankton net 200 µm | Plankton net | https://www.seanoe.org/data/00752/86417/ | Zooplankton abundance and diversity (vertical nets) | Fortier L. |
| *Zooplankton (taxonomy)* | Microscopy | Drifting sediment trap | https://www.seanoe.org/data/00752/86417/ | drifting traps data (25m depth) | Lalande C. |
| *Zooplankton fecal pellets* | Microscopy | Drifting sediment trap | https://www.seanoe.org/data/00752/86417/ | drifting traps data (25m depth) | Lalande C. |

**Table 4. Name and description of variables systematically included in datasets (i.e., in each .csv file)**

| Variable | Description |
|---|---|
| *mission* | mission identifier ("ice_camp_2016", "Amundsen_2016", …) |
| *date* | sampling date (UTC) |
| *latitude* | latitude of sampling location (degree north decimals) |
| *longitude* | longitude of sampling (degree east decimals) |
| *station* | station name (ex: G206) |
| *operation code* | GE_0008-1 (unique operation identifier) |
| *PI* | name(s) of the principal investigator(s) responsible of the measured (calculated) variable |

**Table 5. List of all peer-reviewed journal articles published so far using Green Edge cruise and/or Green Edge Ice Camp data. Only the first 4 authors are indicated in this table.**

| AUTHORS | YEAR | TITLE | JOURNAL | DOI | CRUISE | ICE CAMP |
|---|---|---|---|---|---|---|
| *Amiraux, R., Jeanthon, C., Vaultier, F., Rontani, J.-F.* | 2016 | Paradoxical effects of temperature and solar irradiance on the photodegradation state of killed phytoplankton | Journal of phycology | https://doi.org/10.1111/jpy.12410 | n | y |
| *Rontani, J.-F., Belt, S.-T., Brown, T.-A., Amiraux, R., et al.* | 2016 | Monitoring abiotic degradation in sinking versus suspended Arctic sea-ice algae during a spring ice melt using specific lipid oxidation tracers | Organic Geochemistry | https://doi.org/10.1016/j.orggeochem.2016.05.016 | n | y |
| *Amiraux, R., Belt, S.-T., Vaultier, F., Galindo, V., et al.* | 2017 | Monitoring photo-oxidative and salinity-induced bacterial stress in the Canadian Arctic using specific lipid tracers | Marine chemistry | https://doi.org/10.1016/j.marchem.2017.05.006 | n | y |
| *Rontani, J.-F., Galeron, M.-A., Amiraux, R., Artigue, L., et al.* | 2017 | Identification of di- and triterpenoid lipid tracers confirms the significant role of autoxidation in the degradation of terrestrial vascular plant material in the Canadian Arctic | Organic Geochemistry | https://doi.org/10.1016/j.orggeochem.2017.03.011 | n | y |
| *Dadaglio, L., Dinasquet, J., Obernosterer, I., Joux, F.* | 2018 | Differential responses of bacteria to diatom-derived dissolved organic matter in the Arctic Ocean | Aquatic Microbial Ecology | https://doi.org/10.3354/ame01883 | y | n |
| *Goyens, C., Marty, S., Leymarie, E., Antoine, D., et al.* | 2018 | High Angular Resolution Measurements of the Anisotropy of Reflectance of Sea Ice and Snow | Earth and Space Science | https://doi.org/10.1002/2017EA000332 | n | y |
| *Massicotte, P., Bécu, G., Lambert-Girard, S., Leymarie, E., et al.* | 2018 | Estimating underwater light regime under spatially heterogeneous sea ice in the Arctic | Applied Sciences | https://doi.org/10.3390/app8122693 | n | y |
| *Rontani, J.-F., Amiraux, R., Lalande, C., Babin, M et al.* | 2018 | Use of palmitoleic acid and its oxidation products for monitoring the degradation of ice algae in Arctic waters and bottom sediments | Organic geochemistry | https://doi.org/10.1016/j.orggeochem.2018.06.002 | n | y |
| *Rontani, J.-F., Belt, S.-T., Amiraux, R.* | 2018 | Biotic and abiotic degradation of the sea ice diatom biomarker IP 25 and selected algal sterols in near-surface Arctic sediments | Organic Geochemistry | https://doi.org/10.1016/j.orggeochem.2018.01.003 | n | y |
| *Lafond, A., Leblanc, K., Quéguiner, B., Moriceau, B., et al.* | 2019 | Late spring bloom development of pelagic diatoms in Baffin Bay | Elementa | http://doi.org/10.1525/elementa.382 | y | n |
| *LeBlanc, M., Gauthier, S., Garbus, S. E., Mosbech, A., et al.* | 2019 | The co-distribution of Arctic cod and its seabird predators across the marginal ice zone in Baffin Bay | Elementa | http://doi.org/10.1525/elementa.339 | y | n |

| AUTHORS | YEAR | TITLE | JOURNAL | DOI | CRUISE | ICE CAMP |
|---|---|---|---|---|---|---|
| *Randelhoff, A., Oziel, L., Massicotte, P., Bécu, G., et al.* | 2019 | The evolution of light and vertical mixing across a phytoplankton ice-edge bloom. | Elementa | https://doi.org/10.1525/elementa.357 | y | n |
| *Amiraux, R., Smik, L., Köseoğlu, D., Rontani, J.-F., et al.* | 2019 | Temporal evolution of IP25 and other highly branched isoprenoid lipids in sea ice and the underlying water column during an Arctic melting season. | Elementa | https://doi.org/10.1525/elementa.377 | n | y |
| *Else, B. G. T. T., Whitehead, J. J., Galindo, V., Ferland, J., Mundy, C. J., Gonski, S. F., et al.* | 2019 | Response of the Arctic marine inorganic carbon system to ice algae and under-ice phytoplankton blooms: A case study along the fast-ice edge of Baffin Bay. | Journal of Geophysical Research - Oceans | https://doi.org/10.1029/2018JC013899 | n | y |
| *Gourdal, M., Crabeck, O., Lizotte, M., Galindo, V., et al.* | 2019 | Upward transport of bottom-ice dimethyl sulfide during advanced melting of arctic first-year sea ice. | Elementa | https://doi.org/10.1525/elementa.370 | n | y |
| *Matthes, L. C., Ehn, J. K., L.-Girard, S., Pogorzelec, N. M., et al.* | 2019 | Average cosine coefficient and spectral distribution of the light field under sea ice: Implications for primary production. | Elementa | https://doi.org/10.1525/elementa.363 | n | y |
| *Oziel, L., Massicotte, P., Randelhoff, A., Ferland, J et al.* | 2019 | Environmental factors influencing the seasonal dynamics of spring algal blooms in and beneath sea ice in western Baffin Bay. | Elementa | https://doi.org/10.1525/elementa.372 | n | y |
| *Sampei, M.* | 2019 | An estimation of the quantitative impacts of copepod grazing on an under sea-ice spring phytoplankton bloom in western Baffin Bay, Canadian Arctic | Elementa | https://doi.org/10.1525/elementa.2019.00092 | n | y |
| *Sansoulet, J., Pangrazi, J.-J., Sardet, N., Mirshak, S., et al.* | 2019 | Green Edge Outreach Project: a large-scale public outreach and educational initiative | polar record | https://doi.org/10.1017/S0032247419000123 | n | y |
| *Burgers, T. M., Tremblay, J.-É., Else, B. G. T., & Papakyriakou, T. N.* | 2020 | Estimates of net community production from multiple approaches surrounding the spring ice-edge bloom in Baffin Bay | Elementa | https://doi.org/10.1525/elementa.013 | y | n |
| *Randelhoff, A., Lacour, L., Marec, C., Leymarie, E., et al.* | 2020 | Arctic mid-winter phytoplankton growth revealed by autonomous profilers | Science Advances | https://doi.org/10.1126/sciadv.abc2678 | y | n |
| *Saint-Béat, B., Fath, B. D., Aubry, C., Colombet, J., et al.* | 2020 | Contrasting pelagic ecosystem functioning in eastern and western Baffin Bay revealed by trophic network modeling | Elementa | https://doi.org/10.1525/elementa.397 | y | n |
| *Yunda-Guarin, G., Brown, T. A., Michel, L. N., Saint-Béat, B., et al.* | 2020 | Reliance of deep-sea benthic macrofauna on ice-derived organic matter highlighted by multiple trophic markers during spring in Baffin Bay, Canadian Arctic | Elementa | https://doi.org/10.1525/elementa.2020.047 | y | n |

| AUTHORS | YEAR | TITLE | JOURNAL | DOI | CRUISE | ICE CAMP |
|---|---|---|---|---|---|---|
| *Amiraux, R., Burot, C., Bonin, P., Massé, G., et al.* | 2020 | Stress factors resulting from the Arctic vernal sea ice melt: impact on the viability of the bacterial communities associated to sympagic algae | Elementa | https://doi.org/10.1525/elementa.076 | n | y |
| *Else, B. G. T. T., Whitehead, J. J., Galindo, V., Ferland, J., et al.* | 2020 | Green Edge ice camp campaigns: understanding the processes controlling the under-ice Arctic phytoplankton spring bloom. | Earth System Science Data | https://doi.org/10.5194/essd-12-151-2020 | n | y |
| *Matthes, L. C., Mundy, C. J., L.-Girard, S., Babin, M et al.* | 2020 | Spatial Heterogeneity as a Key Variable Influencing Spring-Summer Progression in UVR and PAR Transmission Through Arctic Sea Ice | Frontiers in Marine Science | https://doi.org/10.3389/fmars.2020.00183 | n | y |
| *Sansoulet, J., Therrien, M., Delgove, J., Pouxviel, G., et al.* | 2020 | An update on Inuit perceptions of their changing environment, Qikiqtaaluk (Baffin Island, Nunavut) | Elementa | https://doi.org/10.1525/elementa.025 | n | y |
| *Ardyna, M., & Arrigo, K. R.* | 2020 | Phytoplankton dynamics in a changing Arctic Ocean. | Nature Clim. Change | https://doi.org/10.1038/s41558-020-0905-y | y | y |
| *Ardyna, M., Mundy, C. J., Mills, M. M., Oziel, L., et al.* | 2020 | Environmental drivers of under-ice phytoplankton bloom dynamics in the Arctic Ocean | Elementa | http://doi.org/10.1525/elementa.430 | y | y |
| *Gérikas Ribeiro, C, dos Santos, A. L., Probert, I., Vaulot, D., et al.* | 2020 | Taxonomic reassignment of *Pseudohaptolina birgeri comb. nov.* (Haptophyta) | Journal of Phycology | https://doi.org/10.1080/00318884.2020.1830255 | y | y |
| *Ribeiro, C. G., Dos Santos, A. L., Gourvil, P., Le Gall, F., et al.* | 2020 | Culturable diversity of Arctic phytoplankton during pack ice melting. | Elementa | http://doi.org/10.1525/elementa.401 | y | y |
| *Tisserand, L., Dadaglio, L., Intertaglia, L., Catala, P., et al.* | 2020 | Use of organic exudates from two polar diatoms by bacterial isolates from the Arctic Ocean | Phil. Trans. R. Soc. A. | https://doi.org/10.1098/rsta.2019.0356 | y | y |
| *Yau, S., Lopes dos Santos, A., Eikrem, W., Gérikas Ribeiro, C., et al.* | 2020 | *Mantoniella beaufortii* and *Mantoniella baffinensis sp. nov.* (Mamiellales, Mamiellophyceae), two new green algal species from the high arctic | Journal of Phycology | https://doi.org/10.1111/jpy.12932 | y | y |
| *Toullec, J., Moriceau, B., Vincent, D., Guidi, L., et al.* | 2021 | Processes controlling aggregate formation and distribution during the Arctic phytoplankton spring bloom in Baffin Bay | Elementa | https://doi.org/10.1525/elementa.2021.00001 | y | n |
| *Vilgrain, L., Maps, F., Picheral, M., Babin, M., et al.* | 2021 | Trait-based approach on zooplankton in situ images reveals contrasted ecological patterns along ice melt dynamics | Limnology & Oceanography | https://doi.org/10.1002/lno.11672 | y | n |

| AUTHORS | YEAR | TITLE | JOURNAL | DOI | CRUISE | ICE CAMP |
|---------|------|-------|---------|-----|--------|----------|
| *Amiraux, R., Rontani, J.-F., Armougom, F., Frouin, E., et al.* | 2021 | Bacterial diversity and lipid biomarkers in sea ice and sinking particulate organic material during the melt season in the Canadian Arctic | Elementa | https://doi.org/10.1525/elementa.2019.040 | n | y |
| *Galí, M., Lizotte, M., Kieber, D.J., Randelhoff, A., et al.* | 2021 | DMS emissions from the Arctic Ocean marginal ice zone | Elementa | https://doi.org/10.1525/elementa.2020.00113 | y | n |
| *Laliberté, J., Rehm, E., Hamre, B., Goyens, C et al.* | 2022 | A method to derive satellite PAR albedo time series over first-year sea ice in the Arctic Ocean | Elementa | https://doi.org/10.1525/elementa.2020.00080 | n | y |