# Peer review of "The Green Edge cruise: Investigating the Marginal Ice Zone processes during late spring / early summer to understand the fate of the Arctic phytoplankton bloom."

_Earth System Science Data, 2022_

## Author Comment (AC1)

Answers to reviewer #1 (anonymous):

Authors are grateful for such a thorough review of our paper as we believe it will greatly improve the manuscript. We have revised the document and it will be proofread by several colleagues before final submission. Authors contributions have been modified to reflect on who wrote the paper.

Responses to general comments

1. Confusion between cruise and ice-camp datasets and both published papers:

    The first data description paper –published by Massicotte et al. in 2020 on the Green-Edge ice camp [https://doi.org/10.5194/essd-12-151-2020] was accompanied by a first dataset containing all the ice camp data, and some of the Amundsen cruise data (the bulk that was ready and processed at the time) [https://doi.org/10.17882/59892]. The reason for that is that we weren't aware a DOI could not be modified, consequently the plan was to add the rest of the cruise data to the same Massicotte et al. DOI on SEANOE. As soon as I (F. Bruyant) sent the cruise paper for publication to ESSD, I was immediately contacted by editor Dave Carlson about the confusion. We had a meeting (editor D. Carlson, editor K. Elger, P. Massicotte and F. Bruyant) and it was decided that the easiest solution would be to create an entirely separated dataset containing ONLY the cruise data with its own DOI [https://doi.org/10.17882/86417] in order to accompany the present paper [essd-2022-41]. The full dataset of Green Edge ice camp data remains unchanged as mentioned in Editor's comment #1 dated March 8$^{th}$ 2022. Furthermore, the cruise data set was momentarily published on the SEANOE website with a publication date as "2016". It has now been corrected for "2022". We have taken the appropriate steps towards appropriate citations of both datasets and both papers in the new version of the paper. We apologize for the confusion, and we hope it is now clarified.

2. List of publications using the Green Edge datasets:

    This is an excellent suggestion. A table will be added to the revised manuscript.

3. Precision on data location:

    Indeed, not all data is on SEANOE and some data is stored in more than one place. We will add a column to Table 3 to indicate where each parameter is to be found. We will also list all storage addresses in the "data availability statement".

Detailed comments

l.72 Comma moved

l.74 We changed capitalization and verified throughout the document

l.75 Changed to "dataset" (and modified throughout)

l.76 Indeed, we corrected the document to reflect late spring / early summer.

l.77 The sentence now reads: "The dataset is available at https://doi.org/10.17882/86417 (Bruyant et al., 2022)", which is the dataset for the cruise. The list of references has also been modified accordingly.

l.81 Wording of the phrase changed to reflect on seasonnality.

l.100 Space added

l.101 Capitalization modified. CCGS Amundsen cruise season is divided in successive legs of 6 weeks each. Legs are numbered sequentially throughout the season which usually runs from beginning of June to end of October - mid November. Each 6 weeks leg can be splitted in two parts (A and B) for logistics reasons which was the case for the Green Edge cruise. However, that same year Leg 2 started right after Green Edge on July 15th for 6 weeks. For everyone working regularly onboard the Amundsen (which is most of my co-authors), it was easier and far less confusing to stick with the conventional numbering of legs. Also some of us actually took part in Leg 2 that year, which was a totally different program.

l.126 The MVP is equipped with several sensors (CTD, C-Star and fluorometers). The subject of the phrase is "a moving vessel profiler". We have modified the punctuation to clarify.

l.156 Changed to "presented"

l.163 Wording and punctuation modified

l.169 Changed to "e.g."

l.203 Citation and reference list have been corrected.

l.209 Changed to "SBE 911plus". We also checked the rest of the manuscript

l.232 Changed to "in situ"

l.249 Space removed

l.251 We have corrected the sentence and the reference list to cite Mueller et al. (2003) and Mobley (1999).

l.262 Changed to "Sea-Bird SBE 19plus"

l.268 We have verified the entire manuscript for unit format and subscripts as well when needed.

l. 295 THe sentence now reads: "The DNA/RNA co-extraction was carried out using the AllPrep DNA/RNA kit (Qiagen)."

l.309 Elsewhere in the manuscript, stations are named without the letter G. We have adopted this semantic throughout the manuscript and explained that it effectively means Transect #, 1 digit then station #, 2 digits.

l.317 Changed to "Leg 1B"

l.343 We added the definition as "subsurface chlorophyll maximum"

l.391 Style has been made homogeneous

l.400 We have modified the phrasing to ensure more clarity.

l.414 Absolutely, the mention of Leg 1 was confusing. We have made the appropriate modifications

l.420 Agreed. This is unnecessary and the final reference will be added in the table listing all the papers based on our dataset.

l.427 Definition has been added in the text.

l.434 Please see answer to the comment on l.309.

l.557 This dataset is hosted on the polar data catalog (PDC). We contacted them to inquire about the proper way to cite their datasets as it seems that they are not giving DOIs. We will make the necessary changes in accordance with their answer.

l.558 the reference for Grasshoff et al. 1999 has been modified

Comments on Figures

Fig.1 Figure one's resolution is 300 dpi, which is the requirement for publication in ESSD. However, we increased the font size, so labels and captions are easier to read. Panel B has been modified to account for the comment. Floats all lived a different number of days; the previous caption was an artifact of the period shown on the graphic, which is the same for all floats. We believe that indicating the ice-covered area on the map (especially with shading) would make the graph too hard to understand (especially with the 7 different positions of the ice edge to be represented. Instead, we modified the figure caption which now reads "**Colored lines indicate the position of the ice edge (sea ice concentration at 80%) on each transect at date of sampling; ice cover persisted on the western side of Baffin Bay, while the eastern side cleared earlier .**".

Fig.3 and 4 Figures will be redone to reach 300 dpi.

Fig. 5 Yes, it would indeed make sense. However, the availability of the data is "as is", meaning monthly averages. Daily averages are not available, consequently we cannot change the span of the data included in Fig.5. You can find the data here: https://apps.ecmwf.int/datasets/data/interim-full-moda/levtype=sfc/.

Fig.7 Transition from sea-ice covered to open water at a given geographic location is a process that takes a fair amount of time. Certainly, it takes several days for the ice pack to break and disperse, leaving the water covered in pieces of ice and slush. DOW is a calculated parameter describing a continuous lengthy process, and we feel that a change in colormap or an abrupt change at zero would be detrimental to the meaning of the graph. Also, at this stage in the paper, it has been mentioned several times that sea ice is located on the Canadian side and open water on the Greenland side.

Fig.10 Figure will be redone to reach 300 dpi.

Fig. 13 We have modified the figure caption for more clarity and homogeneity with the other figures.

Answers to reviewer #2 (Emilia Trudnowska):

We (the authors) are thankful for this detailed review and comments which will greatly improve the quality of the manuscript.

Title: This is a good point. We propose "The Green Edge cruise: Investigating the Marginal Ice Zone processes during late spring / early summer to understand the fate of the Arctic phytoplankton bloom."

Abstract: We have made changes and added more details about our sampling strategies. Abstract now reads as:

"The Green Edge project was designed to investigate the onset, life and fate of a phytoplankton spring bloom (PSB) in the Arctic Ocean. The lengthening of the ice-free period and the warming of seawater, amongst other factors, have induced major changes in Arctic Ocean biology over the last decades. Because the PSB is at the base of the Arctic Ocean food chain, it is crucial to understand how changes in the Arctic environment will affect it. Green Edge was a large multidisciplinary collaborative project bringing researchers and technicians from 28 different institutions in seven countries together, aiming at understanding these changes and their impacts into the future. The fieldwork for the Green Edge project took place over two years (2015 and 2016) and was carried out from both an ice-camp and a research vessel in Baffin Bay, in the Canadian Arctic. This paper describes the sampling strategy and the dataset obtained from the research cruise, which took place aboard the Canadian Coast Guard Ship (CCGS) Amundsen in late spring/early summer 2016. Sampling strategy was designed around the repetitive perpendicular crossing of the marginal ice zone (MIZ), using –not only ship-based station discrete sampling, but also high-resolution measurements from autonomous platforms (Gliders, BGC-Argo floats…) and under-way monitoring systems. The dataset is available at https://doi.org/10.17882/86417 (Bruyant et al., 2022)."

l.86 Both reviewers made the same comment, and we think this is a great suggestion. We will add a Table containing the list of all papers published using the Green Edge cruise dataset. We added the suggested reference to the text and reference list.

l.119 Thanks. We think OWD is a very useful and powerful tool to decipher the complexity of the MIZ processes.

Table 1– Not mentioning the UVP was definitively a mistake. However, as it was onboard the rosette carousel, we added it in Table 2. LISST was not onboard the Amundsen (it was only on the ice camp). We added the sediment traps in Table 1. This table really was focusing on the systematic ship-based operations, repeatedly executed at EACH one of the station types. Therefore, we did not include in the table the deployments or measurement/equipment used only scarcely during the cruise (e.g. gliders). The complete list of equipment deployed during the cruise is in fact included in Table 3, as the "METHOD" column does list the equipment used. However, to follow the reviewer's recommendation, we did add the list of sensors included in the IOP and AOP packages in Table 1.

l.133-135 We have removed the sentences related to "what" is an Argo float.

l.152 Header has been modified it know reads: "Time-for-space formatting and data quality control"

l.172 Agreed, we changed the title to "Description of data collection".

l.176 & others. The list of parameters contained in Table 3 is organized by alphabetical order. We believe this is the easiest way possible to find each parameter if needed. If numbers were added to each of them, it would be the exact same information with no additional value.

l.389 We have modified the sentence for more clarity, it now reads "in case further analysis are needed in the future."

Comments on Figures

Fig. 6: Indeed, the black line represents the ship route. The caption has been modified to include this information.

Fig. 10: The word "microbial taxa" has been added to the figure caption to improve clarity.

Fig. 15: This is a great suggestion. We believe it will help to understand the complexity of the Arctic Ocean system and help visualize the extent of the Green Edge study. Fig. 15 will be added as panel B to Fig. 2.

Comments about dataset.

1. Over the construction of this publication, we had several troubles due mostly to our lack of experience in data/DOI publication. The first data description paper –published by Massicotte et al. in 2020 on the Green-Edge ice camp [https://doi.org/10.5194/essd-12-151-2020] was accompanied by a first dataset containing all the ice camp data, and some of the Amundsen cruise data (the bulk that was ready and processed at the time) [https://doi.org/10.17882/59892]. The reason for that is that we weren't aware a DOI could not be modified, consequently the plan was

to add the rest of the cruise data to the same Massicotte et al. DOI on SEANOE. As soon as I sent the cruise paper for publication to ESSD, I was immediately contacted by editor Dave Carlson about the confusion. We had a meeting (editor D. Carlson, editor K. Elger, P. Massicotte and F. Bruyant) and it was decided that the easiest solution would be to create an entirely separated dataset containing ONLY the cruise data with its own DOI [https://doi.org/10.17882/86417] in order to accompany the present paper [essd-2022-41]. The full dataset of Green Edge ice camp data remains unchanged as mentioned in Editor's comment 1 dated March 8th, 2022. Furthermore, the cruise data set was momentarily published on the SEANOE website with a publication date as "2016". It has now been corrected for "2022". We have taken the appropriate steps towards appropriate citations of both datasets and both papers in the new version of the paper. We apologize for the confusion, and we hope it is now clarified. In the meantime, I have requested the LEFE CYBER website to host the entire raw data of both Green Edge Ice-Camp and cruise, be accessible without password. We will also –as stated in the response to reviewer #1 add a column to Table 3 to indicate where each parameter is to be found. We will also list all storage addresses in the "data availability statement".

---

## Author Response (AR1)

Answers to reviewer #1 (anonymous):

Authors are grateful for such a thorough review of our paper as we believe it will greatly improve the manuscript. We have revised the document and it will be proofread by several colleagues before final submission. Authors contribution has been modified to reflect on who wrote the paper.

Responses to general comments

1. Confusion between cruise and ice-camp datasets and both published papers:

    The first data description paper –published by Massicotte et al. in 2020 on the Green-Edge ice camp [https://doi.org/10.5194/essd-12-151-2020] was accompanied by a first dataset containing all the ice camp data, and some of the Amundsen cruise data (the bulk that was ready and processed at the time) [https://doi.org/10.17882/59892]. The reason for that is that we weren't aware a DOI could not be modified, consequently the plan was to add the rest of the cruise data to the same Massicotte et al. DOI on SEANOE. As soon as I (F. Bruyant) sent the cruise paper for publication to ESSD, I was immediately contacted by editor Dave Carlson about the confusion. We had a meeting (editor D. Carlson, editor K. Elger, P. Massicotte and F. Bruyant) and it was decided that the easiest solution would be to create an entirely separated dataset containing ONLY the cruise data with its own DOI [https://doi.org/10.17882/86417] in order to accompany the present paper [essd-2022-41]. The full dataset of Green Edge ice camp data remains unchanged as mentioned in Editor's comment #1 dated March 8$^{th}$ 2022. Furthermore, the cruise data set was momentarily published on the SEANOE website with a publication date as "2016". It has now been corrected for "2022". We have taken the appropriate steps towards appropriate citations of both datasets and both papers in the new version of the paper. We apologize for the confusion, and we hope it is now clarified.

2. List of publications using the Green Edge datasets:

    This is an excellent suggestion. A table has been added to the revised manuscript. Please see Table 5 on page 55.

3. Precision on data location:

    Indeed, not all data is on SEANOE and some data is stored in more than one place. We added two columns to Table 3 to indicate where each parameter is to be found. One column ("access to dataset") is a direct link to the storage location and another column

("name of file") is the name of the datafile if applicable. We will also list all storage addresses in the "data availability statement" section (now on page 15).

Detailed comments

l.72 Comma moved (now line 79)

l.74 We changed capitalization and verified throughout the document (now line 81)

l.75 Changed to "dataset" (and modified throughout) (now line 81)

l.76 Indeed, we corrected the document to reflect late spring / early summer (now line 83)

l.77 The sentence now reads: "The dataset is available at https://doi.org/10.17882/86417 (Bruyant et al., 2022)", which is the dataset for the cruise (line 86). The list of references has also been modified accordingly.

l.81 Wording of the phrase changed to reflect on seasonality (now line 90)

l.100 Space added (now line 112)

l.101 Capitalization modified (now line 112). CCGS *Amundsen* cruise season is divided in successive legs of 6 weeks each. Legs are numbered sequentially throughout the season which usually runs from beginning of June to end of October - mid November. Each 6 weeks leg can be split in two parts (A and B) for logistics reasons which was the case for the Green Edge cruise. However, that same year Leg 2 started right after Green Edge on July 15th for 6 weeks. For everyone working regularly onboard the Amundsen (which is most of my co-authors, it was easier and far less confusing to stick with the conventional numbering of legs. Also, some of us took part in Leg 2 that year, which was a totally different program.

l.126 The MVP is equipped with several sensors (CTD, C-Star and fluorometers). The subject of the phrase is "a moving vessel profiler". We have modified the punctuation to clarify (now line 138).

l.156 Changed to "presented" (now line 170).

l.163 Wording and punctuation modified (now line 179).

l.169 Changed to "e.g." (now line 184).

l.203 Citation and reference list have been corrected (now line 220-221).

l.209 Changed to "SBE 911plus". We also checked the rest of the manuscript (now line 227).

l.232 Changed to "in situ" (now line 251).

l.249 Space added (now line 268).

l.251 We have corrected the sentence and the reference list to cite Mueller et al. (2003) and Mobley (1999) (now line 271).

l.262 Changed to "Sea-Bird SBE 19plus" (now line 282).

l.268 We have verified the entire manuscript for unit format and subscripts as well when needed (now line 289).

l.295 The sentence now reads: "The DNA/RNA co-extraction was carried out using the AllPrep DNA/RNA kit (Qiagen)." (now line 317).

l.309 Elsewhere in the manuscript, stations are named without the letter G. We have adopted this semantic throughout the manuscript and explained that it effectively means Transect #, 1 digit then station #, 2 digits (now line 330).

l.317 Changed to "Leg 1B" (now line 340).

l.343 We added the definition as "subsurface chlorophyll maximum" (now line 368).

l.391 Style has been made homogeneous (now line 420).

l.400 We have modified the phrasing to ensure more clarity (now line 429).

l.414 Absolutely, the mention of Leg 1 was confusing. We have made the appropriate modifications (now line 443).

l.420 Agreed. This is unnecessary and the final reference will be added in the table listing all the papers based on our dataset (now line 451).

l.427 Definition has been added in the text (now line 458).

l.434  please see answer to the comment on l.309 (now line 465).

l.577 This dataset is hosted on the polar data catalog (PDC). We contacted them to inquire about the proper way to cite their datasets as it seems that they are not giving DOIs. We added a link in Table 3 to the data and removed the reference Guillot et al. 2016 (now line 654).

l.558 the reference for Grasshoff et al. 1999 has been modified (now line 652).

Comments on Figures

Fig.1 Figure one's resolution is 300 dpi, which is the requirement for publication in ESSD. However, we increased the font size, so labels and captions are easier to read. Panel B has been modified to account for the comment. Floats all lived a different number of days; the previous caption was an artifact of the period shown on the graphic, which is the same for all floats. We believe that indicating the ice-covered area on the map (especially with shading) would make the graph too hard to understand (especially with the 7 different positions of the ice edge to be represented. Instead, we modified the figure caption which now reads "**Colored lines indicate the position of the ice edge (sea ice concentration at 80%) on each transect at date of sampling; ice cover persisted on the western side of Baffin Bay, while the eastern side cleared earlier .**" (see figure 1 page 23)

Fig.3 and 4 Figures will be redone to reach 300 dpi (see figures 2 and 3 on pages 24 and 25).

Fig. 5 Yes, it would indeed make sense. However, the availability of the data is "as is", meaning monthly averages. Daily averages are not available, consequently we cannot change the span of the data included in Fig.5. You can find the data here: https://apps.ecmwf.int/datasets/data/interim-full-moda/levtype=sfc/.

Fig.7 Transition from sea-ice covered to open water at a given geographic location is a process that takes a fair amount of time. Certainly, it takes several days for the ice pack to break and disperse, leaving the water covered in pieces of ice and slush. DOW is a calculated parameter describing a continuous lengthy process, and we feel that a change in colormap or an abrupt change at zero would be detrimental to the meaning of the graph. Also, at this stage in the paper, it has been mentioned several times that sea ice is located on the Canadian side and open water on the Greenland side.

Fig.10 Figure will be redone to reach 300 dpi (see new figure 10 on page 32).

Fig. 13 We have modified the figure caption for more clarity and homogeneity with the other figures (now on page 35).

We (the authors) are thankful for this detailed review and comments which will greatly improve the quality of the manuscript.

Title: This is a good point. We propose "The Green Edge cruise: Investigating the Marginal Ice Zone processes during late spring / early summer to understand the fate of the Arctic phytoplankton bloom." (see now title on page 1)

Abstract: We have made changes and added more details about our sampling strategies. Abstract now reads as:

"The Green Edge project was designed to investigate the onset, life, and fate of a phytoplankton spring bloom (PSB) in the Arctic Ocean. The lengthening of the ice-free period and the warming of seawater, amongst other factors, have induced major changes in Arctic Ocean biology over the last decades. Because the PSB is at the base of the Arctic Ocean food chain, it is crucial to understand how changes in the Arctic environment will affect it. Green Edge was a large multidisciplinary collaborative project bringing researchers and technicians from 28 different institutions in seven countries together, aiming at understanding these changes and their impacts into the future. The fieldwork for the Green Edge project took place over two years (2015 and 2016) and was carried out from both an ice-camp and a research vessel in Baffin Bay, in the Canadian Arctic. This paper describes the sampling strategy and the dataset obtained from the research cruise, which took place aboard the Canadian Coast Guard Ship (CCGS) Amundsen in late spring/early summer 2016. Sampling strategy was designed around the repetitive perpendicular crossing of the marginal ice zone (MIZ), using –not only ship-based station discrete sampling, but also high-resolution measurements from autonomous platforms (Gliders, BGC-Argo floats…) and under-way monitoring systems. The dataset is available at https://doi.org/10.17882/86417 (Bruyant et al., 2022)."

l.86 Both reviewers made the same comment, and we think this is a great suggestion. We added a Table containing the list of all papers published using the Green Edge cruise dataset. We added the suggested reference to the text and reference list (see Table 5 on page 55).

l.119 Thanks. We think OWD is a very useful and powerful tool to decipher the complexity of the MIZ processes.

Table 1– Not mentioning the UVP was definitively a mistake. However, as it was onboard the rosette carousel, we added it in Table 2. LISST was not onboard the Amundsen (it was only on the ice camp). We added the sediment traps in Table 1. This table really was focusing on the systematic ship-based operations, repeatedly executed at EACH one of the station types. Therefore, we did not include in the table the deployments or measurement/equipment used only scarcely during the cruise (e.g., gliders). The complete list of equipment deployed during the cruise is in fact included in Table 3, as the "METHOD" column does list the equipment used. However, to follow the reviewer's recommendation, we did add the list of sensors included in the IOP and AOP packages in Table 1.

l.133-135 We have removed the sentences related to "what" is an Argo float (now lines 1144 to 148).

l.152 Header has been modified it know reads: "Time-for-space formatting and data quality control" (now line 166).

l.172 Agreed, we changed the title to "Description of data collection" (now line 186).

l.176 & others. The list of parameters contained in Table 3 is organized by alphabetical order. We believe this is the easiest way possible to find each parameter if needed. If numbers were added to each of them, it would be the exact same information with no additional value.

l.389 We have modified the sentence for more clarity, it now reads "in case further analysis are needed in the future." (now line 417).

Comments on Figures

Fig. 6: Indeed, the black line represents the ship route. The caption has been modified to include this information (see page 28).

Fig. 10: The word "microbial taxa" has been added to the figure caption to improve clarity (see page 32).

Fig. 15: This is a great suggestion. We believe it will help to understand the complexity of the Arctic Ocean system and help visualize the extent of the Green Edge study. Fig. 15 will be added as panel B to Fig. 2 (which is now figure 4 see page 26).

Comments about dataset.

1. Over the construction of this publication, we had several troubles due mostly to our lack of experience in data/DOI publication. The first data description paper –published by Massicotte et al. in 2020 on the Green-Edge ice camp [https://doi.org/10.5194/essd-12-151-2020] was

accompanied by a first dataset containing all the ice camp data, and some of the Amundsen cruise data (the bulk that was ready and processed at the time) [https://doi.org/10.17882/59892]. The reason for that is that we weren't aware a DOI could not be modified, consequently the plan was to add the rest of the cruise data to the same Massicotte et al. DOI on SEANOE. As soon as I sent the cruise paper for publication to ESSD, I was immediately contacted by editor Dave Carlson about the confusion. We had a meeting (editor D. Carlson, editor K. Elger, P. Massicotte and F. Bruyant) and it was decided that the easiest solution would be to create an entirely separated dataset containing ONLY the cruise data with its own DOI [https://doi.org/10.17882/86417] in order to accompany the present paper [essd-2022-41]. The full dataset of Green Edge ice camp data remains unchanged as mentioned in Editor's comment 1 dated March 8[th] 2022. Furthermore, the cruise data set was momentarily published on the SEANOE website with a publication date as "2016". It has now been corrected for "2022". We have taken the appropriate steps towards appropriate citations of both datasets and both papers in the new version of the paper. We apologize for the confusion, and we hope it is now clarified. In the meantime, I have requested the LEFE CYBER website to host the entire raw data of both Green Edge Ice-Camp and cruise, be accessible without password. We will also –as stated in the response to reviewer #1 add a column to Table 3 to indicate where each parameter is to be found. We will also list all storage addresses in the "data availability statement" section.

---

## Referee Report (RR1)

I greatly appreciate the quality of the corrections made in a manuscript by the authors and I am happy that some of my suggestions were found useful. In general, a presented article describes the dataset and the project in concise, interesting and satisfactory manner, in my opinion the provided description should be sufficient for external users to understand the data design, value and potential.

However, even though it was 'promised' in a response to reviewers, to clarify a relation to the first data description paper –published by Massicotte et al. in 2020, I could not find it in a revised version. A clear explanation and reference to the content and difference with this data description paper should be added to avoid further confusion.

Table 3 is a great effort to make it easier to localize specific data, but the provided http links do not work. Moreover, the authors did not comment on the mistakes pointed by me within some, exemplary datasets. I hope they have been fixed, wherever possible.

Some technical details should be double-checked/verified by specific data owners to eliminate potential technical errors, e.g.:
- a unit at Figure 11 – is it possible that biovolume of copepods was 8000 cm3/m3.
- Table 2 – there is some confusion between the title and content, as the header for the 1$^{st}$ column is "Sensor", so it should not be 'particles', but UVP5. It could be "Particles" or rather "particles & plankton", if the Table contained also a column with 'parameter' or 'Variable'

---

## Author Response (AR2)

Response to reviewer Emilia Trudnowska - report-1:

The authors are grateful for all the comments and changes provided by both reviewers in Round 1 and now for round 2, and we are pleased to see our modifications improved the manuscript.

The confusion –pointed out by both reviewers on round 1, concerning the relation between both data papers, was mostly since –at that time, there was only one dataset published. Since then, another dataset has been published under its own DOI for the cruise data. We genuinely thought that having now clearly two separate papers and two datasets (each with its own DOI) would be clear enough for the readers (who would all be unaware of the initial problem). That is the reason why we did not add any explanation in the manuscript for the last round. However, we have now added a new paragraph at the end of the introduction section to make sure no confusion is possible.

Table 3 links work fine on the .docx version of the file, but for some reasons the links to the SEANOE website do not function on the pdf version. Apparently, this is due to the link not being on a single lane. There is nothing we can do but try to have the table formatted properly but the problem may reappear when the manuscript will be published electronically. In the meantime, we have made the link column larger to avoid the problem. I will make the publisher aware of the problem. We also noted that access to the links http://www.obs-vlfr.fr/proof/php/GREENEDGE/ were still restricted by a login/password combination. We have made the necessary request for the protection to be removed and the data to become public. We are awaiting the confirmation anytime soon.

All the very specific mistakes pointed out during round one, were about data published in the original dataset (https://www.seanoe.org/data/00487/59892/, which is mostly Ice-Camp data). Since we created a new dataset exclusive to the cruise data, it was no longer our goal to correct any problem pertaining to the Ice-Camp dataset. However, we took great care in double-checking the files uploaded for the cruise dataset again. Please also note that being a published DOI, the files cannot be changed anymore.

Units on Figure 11: Indeed 8000 $cm^3/m^3$ is an exceptionally high value for non-integrated Copepod biomass. We went back to the data files and to the data owner and verified all calculations made to generate this graph. It is legitimate.

Table 2 has been extensively modified to ensure homogeneity and precision within the different columns. Column 1 now read "Parameter" and the sensors are now specifically placed in Column 2. Sensor's makes and models have been ordered within their column and verified throughout the table.